# Quantifying effects of long-range transport of NO$_2$ over Delhi using back-trajectories and satellite data

Ailish M. Graham[1,2], Richard J. Pope[1,2], Martyn P. Chipperfield[1,2], Sandip S. Dhomse[1,2], Matilda Pimlott[1], Wuhu Feng[1,3], Vikas Singh[4], Ying Chen[5,6], Oliver Wild[7], Ranjeet Sokhi[8], Gufran Beig[9]

[1]School of Earth and Environment, University of Leeds, Leeds, LS2 9JT, UK
[2]National Centre for Earth Observation, University of Leeds, Leeds, LS2 9JT, UK
[3]National Centre for Atmospheric Science, University of Leeds, Leeds, LS2 9PH, UK
[4]National Atmospheric Research Laboratory, Gadanki, India
[5]Laboratory of Atmospheric Chemistry, Paul Scherrer Institute, 5232 Villigen PSI, Switzerland
[6]School of Geography, Earth and Environmental Sciences, University of Birmingham, Birmingham, B15 2TT, UK
[7]Lancaster Environment Centre, Lancaster University, Lancaster, LA1 4QY, UK
[8]Centre for Climate Change Research (C3R), School of Physics, Engineering and Computer Science, University of Hertfordshire, Hatfield, AL10 9AB, UK
[9]National Institute of Advanced Studies, Indian Institute of Science Campus, Bangalore-560012, India

*Correspondence to*: Ailish M. Graham (A.M.Graham@leeds.ac.uk)

**Abstract.**

Exposure to air pollution is a leading public health risk factor in India, especially over densely populated Delhi and the surrounding Indo-Gangetic Plain. During the post-monsoon months, the prevailing north-westerly winds are known to influence aerosol pollution events in Delhi, by advecting pollutants from agricultural fires as well as from local sources. Here we investigate the year-round impact of meteorology on gaseous nitrogen oxides (NO$_x$ = NO + NO$_2$). We use bottom-up NO$_x$ emission inventories (anthropogenic and fire) and high-resolution satellite measurement based tropospheric column NO$_2$ (TCNO$_2$) data, from S5P on-board TROPOMI, alongside a back-trajectory model (ROTRAJ) to investigate the balance of local and external sources influencing air pollution changes in Delhi, with a focus on different emission sectors. Our analysis shows that accumulated emissions (i.e. integrated along the trajectory path, allowing for chemical loss) are highest under westerly, north-westerly and northerly flow during pre- (February - May) and post- (October - February) monsoon periods. According to this analysis, during the pre-monsoon period, the highest accumulated satellite TCNO$_2$ trajectories come from the east and north-west of Delhi. TCNO$_2$ is elevated within Delhi and the Indo-Gangeatic Plain (IGP) to the east of city. The accumulated NO$_x$ emission trajectories indicate that the transport and industry sectors together account for more than 80% of the total accumulated emissions, which are dominated by local sources (>70%) under easterly winds and north-westerly winds. The high accumulated emissions estimated during the pre-monsoon season under north-westerly wind directions are likely to be driven by high NO$_x$ emissions locally and in nearby regions (since NO$_x$ lifetime is reduced and the boundary layer is relatively deeper in this period). During the post-monsoon period the highest accumulated satellite TCNO$_2$ trajectories are advected from Punjab and Haryana, where satellite TCNO$_2$ is elevated, indicating the potential for the long-range transport of agricultural burning emissions to Delhi. However accumulated NO$_x$ emissions indicate local (70%) emissions from the transport sector are the largest contributor to the total accumulated emissions. High local emissions, coupled with a relatively long NO$_x$ atmospheric lifetime and shallow boundary-layer aid the build-up of emissions locally, and along the trajectory path. This indicates the possibility that fire emissions datasets may not capture emissions from agricultural wate burning in the north-west sufficiently to accurately quantify their influence on Delhi AQ. Analysis of daily ground-based NO$_2$ observations indicates that high pollution episodes (> 90$^{th}$ percentile) occur predominantly in the post-monsoon and more than 75% of high pollution events are

primarily caused by local sources. But there is also a considerable influence from non-local (30%) emissions from the transport sector during these periods. Overall, we find that in the post-monsoon period, there is substantial accumulation of high local NOx emissions from the transport sector (70% of total emissions, 70% local), alongside the import of $NO_x$ pollution into Delhi (30% non-local). This work indicates that both, high local $NO_x$ emissions from the transport sector, and the advection of highly polluted air originating from outside Delhi is of concern for the population. As a result, air quality mitigation strategies need to be adopted not only in Delhi but in the surrounding regions to successfully control this issue. In addition, our analysis suggests that the largest benefits to Delhi $NO_x$ air quality would be seen with targeted reductions in emissions from the transport and agricultural waste burning sectors, particularly during post-monsoon months.

## 1 Introduction

The Indo-Gangetic Plain (IGP) covers ~20% of the Indian subcontinent geographical area and houses almost 40% of the population, producing >40% of the total food grain (Badarinath et al., 2009). Within the region, much of the agricultural land is used for rice and wheat production, with ~12 million ha in a wheat-rice crop pattern (Badarinath et al., 2009). Each year there are two major growing seasons; rice is grown from May-September, with harvesting in October-November, and wheat is grown from November-April, with harvesting in April-May. Since the 1980s there has been a shift towards machine harvesting of crops (Badarinath et al., 2009; Jethva et al., 2019). However, this method leaves a large portion of the crop stem root-bound and difficult to remove quickly before sowing the next crop. Wheat stubble is valued for animal feed and a considerable fraction is removed for this purpose before burning. However, rice stubble is generally burned to clear the land quickly as the high silica content of rice stubble means it's not suitable for animal consumption and burning is both more economically viable and quicker than manual removal (Ahmed et al., 2015). Recently Kumar *et al.* (2021) estimated that a total of ~17,000 Tg of rice stubble is burnt each year in India. As a result, burning of rice stubble in the north-west IGP alone accounts for almost 85% of biomass burning in India seasonally and almost 20% annually (Jethva et al., 2019).

The timing of agricultural waste burning has shifted over the past decade due to the introduction of the Subsoil Water Act (SSWA). The SSWA was passed by the government of Punjab in 2009 and requires farmers to delay sowing and transplantation of rice until mid-May and mid-June, respectively (Singh, 2009). The law was introduced to reduce ground water demand by bringing the rice-growing season into line with the arrival of the summer monsoon rainfall (Singh, 2009). However, the SSWA shortened the time period between rice harvest and wheat sowing, resulting in an increased reliance on rice stubble burning to clear land more quickly (Jethva et al., 2019). This has been observed through shifts in the timing of burning from early to late October and an increase in the prevalence of fires (Sembhi et al., 2020; Liu et al. 2021; Kant et al., 2022;).

During the post-monsoon months of October and November synoptic meteorology is favourable for the advection of air pollution from agricultural burning towards Delhi and surrounding areas (e.g. Jethva et al., 2018; Mor et al., 2022). Bikkina *et al*. (2019) performed cluster analysis of back-trajectories released from Delhi and found that north-west IGP emissions have a large (80-100%) influence on air quality in Delhi during autumn (October–November) and winter months (January–February). Additionally, a shallow boundary layer over Delhi and the surrounding IGP region during autumn and winter months, low wind-speeds (1-3 ms$^{-1}$) and poor ventilation and mixing means air pollutants from the IGP advected towards Delhi remain close to the surface, aiding the build-up of pollutants. Previous studies have found that ~40% of black carbon loadings in Delhi originate from crop residue and other biomass burning sources in October and November (e.g. Bikkina et al., 2019; Kanawade et al., 2020).

Delhi is one of the world's largest cities, with a population exceeding 11 million (Census of India. 2011), experiencing some of the poorest air quality in the world (Pandey et al., 2021). A rapid population growth coinciding with the ever-expanding transportation and city infrastructure has made Delhi one of the most

polluted megacities in India (Singh et al., 2021), and indeed the world (Kumar et al., 2017; Hama et al., 2020), suffering ~10k cases of premature mortality per year caused by air pollution (Chen et al., 2020). Winter air pollutant concentrations are particularly high, with daily mean particulate matter with a diameter less than 2.5 micrometres ($PM_{2.5}$) concentrations of 100-200 μg m$^{-3}$ (Singh et al., 2021), and hourly concentrations peaking at over 1000 μg m$^{-3}$ (Bikkina et al., 2019). The high pollution levels in Delhi are also apparent in long-term satellite observations of tropospheric column $NO_2$ (TCNO$_2$) (Vohra et al., 2021). As a result of the high pollution levels in Delhi several policies have been implemented over the past few decades (Guttikunda et al. (2023)). For example, an odd-even traffic intervention was brought in during winter and summer in 2016, allowing only odd and even number plates to be used on alternate days. However, analysis of surface observational data did not show any clear concentration reduction in $PM_{2.5}$ (Kumar et al., 2017) highlighting one of the key challenges in this region: a lack of understanding surrounding the impacts of emissions from the wider region on air quality in Delhi, which prevents effective policy implementation.

Several previous studies have used back-trajectories and chemical transport models to link high pollutant concentrations in selected cities to the arrival of air masses which had passed over high emission regions (Wehner et al., 2008; Reddington et al., 2014; dos Santos et al., 2021). Reddington *et al.* (2014) found that during high pollution episodes, linked with biomass burning, increased concentrations of related tracer species (e.g. levoglucosan) were observed (Atwood et al., 2013; Engling et al., 2014). Similarly, back-trajectory methods and weather typing have also been applied to assess the contribution of local emissions and long-range transport to regional and city-level air pollution in many European regions ( Pope et al., 2014, 2016; Graham et al., 2020; Stirling et al., 2020). For example, Pope *et al.* (2014) sampled satellite $NO_2$ under different synoptic weather patterns (Lamb weather types, LWTs) to show that UK $NO_2$ is generally increased under winter-time anti-cyclonic conditions through pollutant accumulation. $NO_2$ was also enhanced under south-easterly flow due to long-range transport of pollutants from continental Europe. The wintertime increase was attributed to the combined effect of increased emissions, more stable meteorological conditions and decreased photolysis, allowing accumulation over emission sources. Subsequently Stirling *et al.* (2020) and Graham *et al.* (2020) developed this methodology further, including the accumulation of emissions along the trajectory path by combining back trajectories with bottom-up emissions inventories, to analyse changes in $NO_x$ and $PM_{2.5}$ concentrations. They found that long-range transport of $NO_x$ and $PM_{2.5}$ played a larger role in cities in the south of the UK than the north, with the highest contribution from long-range transport under easterly, south-easterly, and southerly flows.

Here our approach is similar to Stirling *et al.* (2020) and Graham *et al.* (2020). Our method can be split into 3 parts. For the first time we combine satellite tropospheric column $NO_2$ (TCNO$_2$) (top-down) with back trajectories (released from Delhi during 2017 and 2018). Secondly, we combine anthropogenic and fire emissions of $NO_x$ (bottom-up) with back-trajectories, in the same way as Stirling *et al.* (2020) and Graham *et al.* (2020). Both steps allow us to quantify the contribution of local and non-local emissions sources to poor $NO_x$ AQ in Delhi. And thirdly, we exclude sectors from the bottom-up anthropogenic emissions dataset in order to identify key source sectors contributing to poor $NO_x$ AQ in Delhi. This develops the methodology of Stirling *et al.* (2020) and Graham *et al.* (2020) further. Section 2 describes the datasets and method used, Section 3 presents our results and Section 4 summarises the implications of our findings.

## 2. Methods

### 2.1. Anthropogenic $NO_x$ emissions

We created a merged emissions dataset for India containing both anthropogenic and daily fire emissions (Figure 1f). Table 1 summarises the datasets used to generate this dataset.

### 2.1.1. Global Emissions

Global monthly $NO_x$ emissions are from the Emission Database for Global Atmospheric Research with Task Force on Hemispheric Transport of Air Pollution (EDGAR-HTAP2) 2010 dataset (Janssens-Maenhout et al., 2015), which are at 10 km resolution (Figure 1c). A comparison between emissions over Delhi from the SAFAR and EDGAR-HTAP2 emissions datasets is also shown in Figure 1a and 1b. This clearly demonstrates the SAFAR dataset provides more spatial detail than EDGAR-HTAP2. EDGAR-HTAP2 is a global, gridded, air pollution emission inventory compiled using officially reported, national gridded inventories; if national emissions datasets for specific sectors were not available EDGAR v4.3 grid maps are used. The resulting EDGAR-HTAP2 dataset provides monthly and annual emission distribution along with emission factors that are fuel-, technology-, process- and human activity-dependent. Emissions include all anthropogenic emissions except large-scale biomass burning (e.g. wildfires).

### 2.1.2. Delhi Emissions

Monthly anthropogenic $NO_x$ emissions for Delhi are taken from the 2018 System of Air Quality and Weather Forecasting And Research (SAFAR) dataset (Beig, 2018) (Figure 1a). Data for the SAFAR emission inventory were collected during a 37,500-hour campaign and are provided at very high (400 m) resolution over a region covering 70 km × 65 km. Emissions are provided for 26 different source sectors. Emissions are very detailed, for example emissions for transport were calculated using traffic volume collected by click counters across the region. Collected data was then converted into emissions using a Geographical Information System (GIS)-based statistical model. Since SAFAR provides a much more detailed, and up-to-date, inventory of emissions in Delhi we regrid SAFAR emissions to 10 km and then replace all EDGAR-HTAP2 emissions (see 2.1.2 for more details) in Delhi with SAFAR emissions using a simple mask method. We create an empty 10 km global grid and first add SAFAR emissions. Then we add EDGAR-HTAP2 emissions where the grid is still empty (i.e. where no SAFAR emissions were added). The combined dataset is resampled to daily resolution by interpolating monthly values to daily temporal resolution.

### 2.1.3. Fire Emissions

Wildfire $NO_x$ emissions for 2018 are taken from the Fire Inventory from NCAR (FINNv2.5) dataset (Wiedinmyer et al., 2011) at 10 km resolution (Figure 1e) and added to the anthropogenic emissions dataset (to generate Figure 1f). FINNv2.5 combines data from the Moderate Resolution Imaging Spectroradiometer (MODIS) sensors on NASA's Terra and Aqua satellites with Visible Infrared Imaging Radiometer Suite (VIIRS) on Suomi NPP. We chose to use FINNv2.5 because of its improved ability to detect small, low temperature fires which are likely to be important in India (e.g. agricultural burning). Both MODIS and VIIRS use a Thermal Anomalies Product to provide detections of active fires. MODIS detections are provided at 1 km resolution, while VIIRS has a resolution of 375 m. In FINN, fire hotspot detections from MODIS and VIIRS are combined with land cover, biomass consumption estimates and emissions factors to calculate daily fire emissions globally at 1 km resolution. Burned area is assumed to be 1 $km^2$ or 0.14 $km^2$ for each fire identified by MODIS or VIIRS, respectively, and scaled back based on the density of vegetation from the MODIS Vegetation Continuous Fields (VCF) (i.e. if 50% bare = 0.5 $km^2$ or 0.07 $km^2$ burned area). The type of vegetation burned during a detected fire is determined using the MODIS Collection 5 Land Cover Type (LCT). Each fire pixel is assigned to one of 16 possible land cover/land use. The 16 land cover types are then aggregated into eight generic categories to which fuel loadings are applied (Wiedinmyer et al., 2011). Fuel loadings are from Hoelzemann *et al.* (2004) and emissions factors are from Andrae and Merlet (2001), McMeeking (2008) and Akagi *et al.* (2011). FINN includes all emissions from above-ground vegetation but not from the combustion of peat (Kiely et al., 2019). Fire types included are wildfires, prescribed and agricultural burning. However, trash-burning or biofuel use are not included.

The merged emission dataset described above provides the control scenario in which all sectors are included (Figure 1f). Additionally, a further 6 emissions files were generated where individual sectors are excluded (transport, industry, power, residential, other and fires) to quantify their contribution to accumulated $NO_x$ along the trajectory path.

## 2.2. Satellite data

We use tropospheric column nitrogen dioxide ($TCNO_2$) measurements for February 2018 to January 2020 from the TROPOMI instrument on-board ESA's Sentinel-5 Precursor (S5P) satellite that was launched on 13[th] October 2017. TROPOMI is a hyper-spectral nadir-viewing imager with an equator overpass time at ascending node of 13:30. The $NO_2$ columns are derived using TROPOMI's UVIS spectrometer backscattered solar radiation measurements in the 405–465 nm wavelength range (van Geffen et al., 2015, 2019). The swath is divided into 450 individual measurement pixels, which results in a near-nadir resolution of 7.0 km × 3.5 km. The total $NO_2$ slant column density is retrieved from the Level 1b UVIS radiance and solar irradiance spectra using Differential Optical Absorption Spectroscopy (van Geffen et al. 2019). Tropospheric and stratospheric slant column densities are separated from the total slant column using a data assimilation system based on the TM5-MP chemical transport model, after which they are converted into vertical column densities using a look-up table of altitude-dependent air-mass factors. As data was available from February 2018 onwards, we selected the two years (2018-2019) closest to the back-trajectories (2017-2018) and anthropogenic (2010/2018) and fire emissions (2018) for our analysis. We follow the approach of Pope *et al.* (2018) to map TROPOMI $TCNO_2$ data onto a 0.05° × 0.05° grid over India. The approach of Pope *et al.* (2018) uses an oversampling methodology where TROPOMI pixels are sliced into sub-pixels and mapped onto a high-resolution level-3 grid. Individual retrievals are filtered for a geometric cloud fraction < 0.2 and a quality control flag > 75 using all the available daily TROPOMI data for both years.

## 2.3. Back Trajectories

Our approach is to combine back trajectories with top-down satellite $TCNO_2$ or bottom-up $NO_x$ emission estimates to investigate the influence of long-range transport (advection) of $NO_x$ on Delhi AQ under different wind-directions. In the following section we will refer to bottom-up $NO_x$ emissions, however the method is the same when using satellite $TCNO_2$.

We use primary $NO_x$ emissions integrated over air mass back-trajectories to determine the relative influence of direct $NO_x$ emissions on those air masses. Back-trajectories are calculated using the Reading Offline TRAJectory (ROTRAJ) Lagrangian transport model (Methven, 2003). The model uses dynamical fields from ERA-Interim reanalysis from the European Centre for Medium-Range Weather Forecasts (ECMWF) to calculate trajectories at 1.125° horizontal resolution. After a trajectory parcel is released, its location is calculated every 6 hours; for vertical interpolation the model uses cubic Lagrangian interpolation and horizontal fields are calculated using bilinear interpolation. This approach primarily focuses on large-scale advection and does not resolve small-scale sub-grid turbulent transport or convection.

In this study, ROTRAJ back trajectories were released from just above the surface (0.99 sigma level; a terrain following coordinate system where 1 is the surface) in central Delhi at 06:00 UTC (11:30 local time) for the years 2017 and 2018, extending back 4 days in 6-hour time steps. We choose 06:00 UTC (11:30 local time) as this is close to mid-day local time and so matches the time of the satellite overpass time (13:30 local time) most closely. We are restricted to using ROTRAJ back-trajectories for 2017 and 2018, while using satellite data for 2018 and 2019, as the ECMWF ERA-Interim reanalysis dataset was terminated in August 2019 and satellite data is not available until 2018. $NO_x$ emissions were accumulated along each trajectory over 4 days at 15-minute time intervals (interpolated linearly from 6-hourly position output) (Figure 2).

NO$_x$ emissions were only accumulated only if the trajectory path was within the boundary layer (which we determine based on ERA-Interim reanalysis). At each location, we accumulate the entire emission within an emission grid box (TCNO$_2$: ~5 km, emissions: ~10 km) over which the trajectory passes. The surface area of each grid box that the trajectory points passed over is also accumulated over time.

The along-trajectory emission accumulation can be represented by Equation (1):

$$E = \sum_{i=1}^{N} [E_{i-1} + \emptyset_i . \Delta t . \alpha_i] e^{-\Delta t / \tau_i} \tag{1}$$

where N (=384) and E$_0$ =0.0

$E_i$ is accumulated NO$_x$ (kg) at any given point $i$ along the trajectory (with E at point 0 [E$_0$] being equal to 0), $\phi_i$ is the emissions flux of NO$_x$ (kg m$^{-2}$ s$^{-1}$) at point $i$, $\Delta t$ is the 15-minute time step, $\alpha$ at point $i$ is the surface area of the grid box (m$^2$) and $\tau$ at point $i$ is the specified NO$_x$ lifetime ($\tau$). Therefore, $E$ is total accumulated NO$_x$ mass (kg) and $N$ is the number of 15-minute time steps within the 4-day trajectory (384).

To account for chemical loss of NO$_x$ along the trajectory path, the lifetime of NO$_2$ was calculated at each timestep from TOMCAT (Chipperfield, 2006; Monks et al., 2017) 3-D hourly hydroxyl radical (OH), pressure and temperature fields, assuming the main loss pathway in Equation (2), where NO$_2$ is oxidised by OH to form nitric acid (HNO$_3$) (which dominates over photolysis within the boundary layer). The NO$_x$ lifetime ($\tau$) (Equation (3)) was calculated using a temperature and pressure-dependent rate constant ($k$) (IUPAC, 2022). The calculated lifetime ($\tau$) was then applied to the total NO$_x$ accumulated emission in the air parcel in Equation (1).

$$OH + NO_2 + M \rightarrow HNO_3 + M \tag{2}$$
$$\tau = 1 / (k . [OH]) \tag{3}$$

To remove the dependence of the accumulated emissions calculated in Equation (1) on emission grid resolution (since we assumed the air mass has the same width as the emission grid box), the total accumulated NO$_x$ mass ($E$) was divided by accumulated surface area ($S$) and then scaled by $10^9$ to give $E$ units of µg m$^{-2}$. $S$ is given by Equation (4):

$$S = \sum_{i=1}^{N} a_i \tag{4}$$

Emissions accumulated within Delhi (*EDelhi*) were also determined using the same approach, but only implemented when the trajectories enter the Delhi region, which we define using a bounding box (76.8 - 77.5°E, 28.1 - 28.9°N). To derive *EDelhi* in units of µg m$^{-2}$, the accumulated NO$_x$ mass from Delhi was divided by the accumulated surface area (S) over the full trajectory path. The ratio between *EDelhi/E* represents the fractional contribution of Delhi sources towards the total accumulated NO$_x$ emissions.

Finally, the daily (06:00 UTC) total accumulated emission and *EDelhi/E* ratios from all sites were binned by 8 wind directions (north through to north-westerly) based on their end point in relation to Delhi. This methodology provides a powerful tool to identify which flow directions are the most polluted and to derive the proportion of pollutant emissions from long-range transport versus local sources.

## 2.4 Observational Data

Hourly averaged surface $NO_2$ concentration from 36 sites across the Delhi region for 2018 and 2019 (Figure 3) were obtained from the CAAQMS (Continuous Ambient Air Quality Monitoring Stations) portal (https://app.cpcbccr.com/ccr/#/caaqm-dashboard-all/caaqm-landing) of the Central Pollution Control Board (CPCB) of India. The ambient $NO_2$ concentration was measured based on the Gas Phase Chemiluminescence methodology; the technical details of monitoring can be found in CPCB (2019). The data was further quality controlled to remove outliers and missing values (Singh et al., 2020). As TROPOMI has a local overpass time of approximately 13:30, the average $NO_2$ between 13:00-14:00 hours was calculated for each site to represent the daily $NO_2$ corresponding to the overpass of TROPOMI. From this, the daily median $NO_2$ concentration across all sites was calculated to represent daily $NO_2$ levels over Delhi in 2018 and 2019.

## 3. Results

### 3.1 TCNO$_2$ Back-Trajectory Analysis

The contribution of local and non-local sources to $NO_2$ levels in Delhi are controlled by primary emissions, the chemical lifetime of $NO_2$ and advection by meteorology (i.e. accumulation/advection of pollutants under stable conditions or ventilation of source emissions downwind). Figure 4b, c and d show the annual average TCNO$_2$, surface $NO_2$ lifetime and boundary layer height (BLH) over India along with the anomalies for the pre-monsoon (February – May), monsoon (May - October) and post-monsoon (October - February) seasons.

Peak annual TCNO$_2$ ranges between 4000 and 5000 µg m$^{-2}$ over the source regions (e.g. Delhi, Kolkata, Nagpur and industrial sources in the east (approximately 80-85°E, 20-25°N)) (Figure 4b). Smaller urban $NO_2$ hotspots range between 3000 and 4000 µg m$^{-2}$, higher than values across large European hotspots which peak at 3500 µg m$^{-2}$ (Pope et al., 2019). In the seasonal anomaly, the pre-monsoon TCNO$_2$ values increase by approximately 500-1000 µg m$^{-2}$ around Delhi but with similar decreases to the north-west. However, the majority of the country experiences TCNO$_2$ increases greater than 2000 µg m$^{-2}$. During the monsoon season there is a decrease in TCNO$_2$ by 1000-1500 µg m$^{-2}$ over Delhi, which is reflected in most other urban centres and industrial regions. The post-monsoon season experiences the largest degradation in $NO_2$-related air quality as all hotspots increase by >2000 µg m$^{-2}$ (e.g. for Delhi) with peak enhancements over the industrialised region in the east of India and to the north-west of Delhi in Punjab and Haryana, where agricultural burning is common during this time.

The calculated $NO_2$ lifetime (based on modelled OH, temperature and pressure) typically ranges between 1.5 to 15 hours over India (between 20 and 24 hours over Bangladesh and Myanmar) and 8 to 12 hours over Delhi (Figure 4c). During the pre-monsoon, there is little change from the annual average, decreasing by 1 to 2 hours over Delhi. Along the east Indian coastline, there is a general decrease of 3-5 hours. During the monsoon season, there are large reductions in the $NO_2$ lifetime over the northern segment of the domain, decreasing by 5 to 10 hours, propagating southwards to Delhi with a decrease of approximately 3-5 hours. Central India remains similar to the annual average while the east coastline now experiences increases in the $NO_2$ lifetime of 3 to 5 hours. In the post-monsoon, while southern India experiences little change, Delhi and the north of India see an increase in the lifetime by 3-7 hours.

Western India has an annual average BLH ranging between 800 and 1100 m (400-800 m over eastern India), while Delhi BLH is between 400 and 600 m (Figure 4d). In the pre-monsoon, the Indian boundary layer is well ventilated with an increase in BLH by ~200 to 400 m (50-100 m for Delhi). In the monsoon

season, the boundary layer remains well ventilated with enhancements of typically 100 to 200 m (although with some regions of reduction in the BLH), relative to the annual average. In the post-monsoon season, colder temperatures cause country-wide shallowing of the boundary layer by 200 to 400 m, including Delhi, peaking at over 500 m in central India.

During the pre-monsoon and, especially, post-monsoon seasons conditions are favourable for the degradation of $NO_2$ air quality. In the pre- and post-monsoon primary $NO_x$ emissions (e.g. from increased domestic heating, power demands and agriculture burning) are typically larger, there is a longer $NO_2$ lifetime (i.e. less chemical loss) and a shallower boundary layer, trapping emissions over Delhi and the wider IGP. This effect is largest in the post-monsoon, though it is also apparent in the pre-monsoon. In contrast, during the monsoon there are lower primary $NO_x$ emissions, a shorter $NO_2$ lifetime and increased atmospheric ventilation of the boundary layer, all of which yield lower pollution levels during the monsoon season. We have thus demonstrated the seasonal influences on $NO_2$ levels in Delhi and surrounding region but now exploit the back-trajectory integrated emissions methodology (see Section 2) to determine the balance of local versus non-local sources of $NO_x$ to Delhi air quality. Here, we accumulate $TCNO_2$ from TROPOMI under the back trajectories to investigate the seasonal influence of wind direction on the advection of $TCNO_2$ into Delhi. As discussed above, there is a clear seasonal influence of the monsoon circulation on the back-trajectories integrated emission totals.

On average, the back-trajectories arrive into Dehli at the surface from the south-west, east and north-west (Figure 4a). Trajectories from the south-west have the lowest accumulated $TCNO_2$ levels (0-200 µg m$^{-2}$), largely originating over the Arabian Sea (i.e. fewer $NO_x$ sources) before arriving in Delhi. Trajectories originating from east India pass over several large urban centres and industrial regions yielding integrated $TCNO_2$ trajectories with $TCNO_2$ levels that range between 400 to 900 µg m$^{-2}$ before arriving in Delhi. The trajectories from the north-west are the most polluted, with peak $TCNO_2$ levels of >1500 µg m$^{-2}$. When split seasonally, the pre-monsoon shows trajectories approaching Delhi from the north-west (200->1000 µg m$^{-2}$) and east (200-900 µg m$^{-2}$). In the monsoon, cleaner airmasses (0-200 µg m$^{-2}$) originate from the south-west, while many eastern airmasses remain similarly polluted as the pre-monsoon (200-900 µg m$^{-2}$). During the post-monsoon, most trajectories are from the north-west of India and show the largest seasonal pollution levels (i.e. integrated $TCNO_2$ back-trajectories of typically 600 to >1500 µg m$^{-2}$), suggestive that agricultural waste burning in north-west India contributes to poor AQ in Delhi during this time.

## 3.2 $NO_x$ Emissions Back-Trajectory Analysis

We repeated this approach using the bottom-up emissions inventory (Figure 5) finding the results are generally in close agreement to the $TCNO_2$ results, with the highest accumulated emissions being observed in the pre- and post-monsoon. In the pre-monsoon season, the integrated emission back-trajectories range between 100 and 800 µg m$^{-2}$ from both the east and north-west. Note the emission trajectory values are lower than the equivalent $TCNO_2$ values as they are emission fluxes and not the tropospheric integrated column. In terms of altitude, the trajectories range from the near-surface to 400 hPa. At t=96 hours before arrival at Delhi, the median trajectory position is near 800 hPa before descending below 950 hPa at approximately 30 hours from Delhi (t=30 hours). In the vertical distribution, there is no clear link between the integrated emission trajectory values and altitude. During the monsoon season, there are two limbs of integrated emission trajectories with $TCNO_2$ ranging between 0 and 400 µg m$^{-2}$ from the south-west and 300-500 µg m$^{-2}$ from the east. The average trajectory position is lower in altitude, starting at approximately 950 hPa and remaining below this pressure level. There appears to be a split in the trajectory origins and altitudes with south-western trajectories coming from the Arabian Sea near the surface with few emissions sources (airmasses will be influenced to some extent by shipping emissions but the short $NO_2$ lifetime will yield limited impact in Delhi), while more polluted trajectories (i.e. substantial upwind sources) from the east are typically 900-600 hPa in altitude. During the post-monsoon season, the median trajectory altitude is approximately 875 hPa at t=96 hours before arrival in Delhi. For the final 24 hours or so, the trajectories converge on Delhi below 950 hPa. As a result, while all seasons experience trajectories 1-day out from Delhi below 950 hPa, the post-monsoon trajectories are

closer to the surface than the pre-monsoon equivalent throughout the 4 days and are exposed to larger emission fluxes than those during the monsoon. Secondly, as these post-monsoon trajectories originate from the north-west and are trapped in the IGP by the Himalayas, there is the opportunity for re-circulation of the trajectory over up-wind $NO_x$ sources. Overall, the meteorological, chemical and emission factors, all trapped against the Himalayas, in the post-monsoon season, are key for the substantial degradation in the air quality.

The integrated emission trajectories also provide the opportunity to gain better insight about the proportion of local vs. non-local sources (i.e. integrated trajectories over the full 4 days vs. integrated trajectories just over Delhi). Figure 6 shows this, but note we only plot wind directions where the sample size is greater than or equal to 10. We find that during the pre-monsoon season, the grouped integrated emissions trajectories are moderately polluted (south-west, west, and north-west) range between 30 and 120 $\mu$g m$^{-2}$) with peak average north-westerly and easterly values of approximately 300 $\mu$g m$^{-2}$. In terms of local contributions, all flows are dominated by local sources (i.e. 75-100%). During the monsoon, all wind directions (easterly, south-easterly, south-westerly, westerly and north-westerly) have lower integrated emission back-trajectories (integrated $NO_x$ emissions peaking at approximately 200 $\mu$g m$^{-2}$) with 80-95% of emissions coming from local sources. For poor air quality, as discussed above, the post-monsoon season is the most important as integrated emission trajectory values from the north (sample size (n) = 17), north-west (n = 179), north-east (n = 10) and east (n = 21) range between 300 and >400 $\mu$g m$^{-2}$ on average. Despite the larger integrated emissions trajectory median values, the key factor is the local contribution is much lower, ranging from 65-80%. The north-westerly trajectory median accumulated NOx emissions is the largest and most frequent (i.e. 73% of trajectories originate from the north-west) during the post-monsoon season with approximately 35% of emission contributions coming from outside of Delhi. Thus, we find that the advection of highly polluted air originating from both inside and outside Delhi are important contributors to the Delhi population and air quality mitigation strategies need to be adopted not only in Delhi but also in the surrounding regions to successfully control this issue.

### 3.2.1 Contribution of Individual Emissions Sectors

Next, we quantify the influence of individual sectoral emissions to the total accumulated emissions by running the back trajectory analysis without individual emissions source sectors and analysing accumulated emissions (Figure 7, note that only wind-directions with a sample size greater than or equal to 10 are shown). During the pre-monsoon season, accumulated emissions from the transport sector are the dominant contributor (>70%) and, though local sources dominate (>70%), non-local sources are also important (30%) under the most frequent north-westerly and westerly wind directions. Trajectories travel close to the surface, within the boundary layer, in the final 24 hours and the $NO_x$ lifetime is short. This indicates that a large fraction of the total integrated emissions reaching Delhi are likely to have originated in the region surrounding the city, where the $NO_x$ burden from transport is high. Additionally, the contribution of local industrial and other emissions is evident.

During the monsoon season, easterly (easterly and south-easterly) and westerly (south-westerly and north-westerly) winds dominate. Under all directions local accumulated emissions show nearly similar contribution (>90%) for all sectors. Although many trajectories travel from distant regions, remaining close to the surface, there are few sources over the ocean (apart from shipping) and the $NO_x$ lifetime is short, so the advection of accumulated emissions concentration is much lower. Therefore, the small contribution of non-local sources is likely from nearby regions within the polluted IGP. Overall, transport emissions have the highest contribution to overall accumulated emissions (>70%), which is likely driven by transport emissions dominating the $NO_x$ burden in Delhi. In addition, contributions from local power and industrial emissions are also important under easterly and south-easterly (and north-westerly) wind directions with only a small non-local contribution to the overall emissions (<10%) and are likely to originate from the large industrial region to the east of Delhi and from power generation to the north-west.

During the post-monsoon winds are primarily north-westerly and are associated with high contributions from transport emissions (>70% transport, 70% local emissions), with non-negligible contributions from industry, other, power and residential emissions (and negligible contributions from fires). Again, local sources generally dominate for these sectors, except residential and power (85% and 100% non-local), which have the largest probable contribution from within the IGP region surrounding Delhi where residential emissions are high and to the north-west where many power stations are located. This suggests the increase in accumulated $NO_x$ emissions during the post-monsoon is driven by a combination of several factors. First, a longer $NO_x$ lifetime, arising from OH being less abundant (to react with $NO_x$) enables accumulated emissions to be advected over long (and short) distances from high emission regions north-west of Delhi, increasing the contribution of non-local sources to AQ in Delhi. Second, a shallow boundary layer traps emissions close to the surface, allowing local emissions to accumulate, as well as allowing the accumulation of emissions along the trajectory path. Third, an increase in emissions during winter months due to increased demand for heating (residential sector) and power (power sector).

### 3.2.3 High Pollution Events

Finally, we apply the same back-trajectory emissions method to the high pollution days in Delhi to investigate the drivers of high-pollution events in the city. We use daily median ground-based $NO_2$ observations from sites in Delhi and the surrounding region to identify high pollution days, defined as days where daily median $NO_2$ concentrations are above the 90th percentile of daily median $NO_2$ concentrations (>37 μg m$^{-3}$) between 1st January 2017 and 31st December 2018 (Figure 8a). We then subsample the trajectories using these high pollution days and attribute the accumulated emissions to specific emissions sectors.

There is a clear seasonal cycle in $NO_2$ concentrations in Delhi, with maximum $NO_2$ values occurring during pre- and post-monsoon months that peak in December (up to ~60 μg m$^{-3}$), and minima during monsoon months (20 to ~30 μg m$^{-3}$) (Figure 8a). We also find that BLH is inversely related with median $NO_2$ concentrations, indicating a shallow boundary layer (<600 m) during the pre- and post-monsoon and a higher boundary layer (600-1500 m) during the monsoon (Figure 8b). Additionally, the atmospheric lifetime of $NO_x$ is highest in pre- and post-monsoon months (>10-70 hours) and is lowest during the monsoon season (~10 hours (Figure 8c). Finally, $NO_x$ emissions are also highest during the pre- and post-monsoon (~0.05-0.07 kg m$^{-2}$ s$^{-1}$) and lowest during the monsoon season (<0.055 kg m$^{-2}$ s$^{-1}$ Figure 8d). The combination of these factors aids the accumulation of increased local emissions during the pre- and post-monsoon months and the dispersal of the decreased emissions during the monsoon. Thus, the non-monsoon seasons, especially in the post-monsoon, experience a substantial degradation in air quality.

The back trajectory emissions analysis indicates that high $NO_2$ pollution days are associated with trajectories from the north-west of Delhi (Figure 9a) in both the pre- (n=12 days) and post-monsoon (n=60 days). Note that we only include wind-directions with a sample size greater than or equal to 5 (easterly and north-westerly). Trajectories gradually descend towards the surface between 90 and 20 hours out from Delhi, remaining close to the surface (>950 hPa) until they reach Delhi (Figure 9b). However, during the post-monsoon trajectories are much closer to the surface from hour 90 (910-810 hPa) compared with the pre-monsoon (840-770 hPa), likely allowing increased accumulation of emissions. During post-monsoon high-pollution days, local (~70% local) transport emissions dominate (>75%) the total accumulated emissions (Figure 10). This is likely due to the trajectories remaining close to the surface for the final 24 hours, within a shallow boundary layer, at a period when the $NO_x$ atmospheric lifetime and emissions are increased. The contribution of other sectors to the remaining accumulated emissions (residential, industrial, other and power) is smaller but local emissions remain important (25-100%). There is also a negligible contribution (<2 μg m$^{-3}$) from non-local (100%) fires under north-westerly winds. Overall, this suggests that non-local $NO_x$ emissions from long-range transport (advection) contribute to poor $NO_2$ AQ in Delhi but the accumulation of local emissions under a shallow boundary layer dominate.

## 4. Discussion and Conclusions

### 4.1 Seasonal Wind Flow Regimes

In this study we have used high-resolution satellite tropospheric column $NO_2$ ($TCNO_2$) data from TROPOMI and bottom-up $NO_x$ emission inventories alongside a back-trajectory model to investigate the influence of local and the advection of pollutants to Delhi throughout the year. We also performed sensitivity simulations to separate the influence of various sectoral emissions on the accumulated pollution, by removing individual emission source sectors. We summarise our findings in Table 2, which gives the dominant wind-direction(s) in each season as a percentage of days in the season, the source sectors with the largest contribution to the accumulated emissions under that wind direction and finally, the contribution of local (Delhi) or non-local (rest of India) accumulated emissions to the total accumulated emissions. Our analysis indicates that both $TCNO_2$ and emissions accumulated over 4 days are highest under easterly, northerly and north-westerly directions during the pre- ($TCNO_2$: 200-1000 $\mu g$ $m^{-2}$, Emissions: ~300-800 $\mu g$ $m^{-2}$) and post- ($TCNO_2$: 600-1500 $\mu g$ $m^{-2}$, Emissions: ~200->1000 $\mu g$ $m^{-2}$) monsoon months. During the pre-monsoon north-westerly wind directions dominate (54%). The seasonal satellite $TCNO_2$ anomaly indicates decreased $TCNO_2$ to the north-west of Delhi but increased $TCNO_2$ locally in Delhi, suggestive that local emissions are the dominant influence on AQ in Delhi during this time. In agreement with this, accumulated $NO_x$ emissions indicate that emissions from the transport sector account for >70% of the total accumulated emissions but indicate ~70% contribution from local emissions. The high accumulated emissions observed under these wind directions are likely driven by high local $NO_x$ emissions during the pre-monsoon (since $NO_x$ lifetime is relatively short and the boundary layer is relatively deep). During the post-monsoon, north-westerly wind dominates (73%). The seasonal satellite $TCNO_2$ anomaly indicates increased $TCNO_2$ over Haryana and Punjab and Delhi, suggestive that agricultural waste burning and local emissions are likely to influence AQ in Delhi during this time. In contrast, accumulated $NO_x$ emissions indicate that transport emissions (>70%), which are predominantly local (70%), are most important. During the post-monsoon period high emissions are accompanied by a longer $NO_x$ atmospheric lifetime and shallow boundary-layer. These two mechanisms aid the build-up of emissions along the trajectory path and lead to increased advection of $NO_x$ into the city under north-westerly flow. In addition, the accumulation of local emissions is increases too. Alongside this, local and non-local (50-100%) residential, other and industrial emissions also contribute to the total accumulated emissions under north-westerly winds during the post-monsoon. Emissions are likely due to increased demand for heating and energy during the colder post-monsoon months in nearby states, such as Uttar Pradesh, where there is a strong reliance on open stoves and fires for cooking and heating.

It should be noted that the mismatch between the spatial pattern of $TCNO_2$ anomalies, which clearly indicate increased $TCNO_2$ over agricultural waste burning regions in the post-monsoon, and the $NO_x$ emissions sectoral analysis may suggest fire emissions are underestimated in the fire emissions dataset. The reasons for this are discussed further in section 4.3.

### 4.2 Source Attribution during High Pollution Episodes

We also applied the back-trajectory method to high-pollution days, defined as days where median $NO_2$ concentrations (from ground-based observations) are above the 90[th] percentile. The ground-based observations indicate that high-pollution episodes are most common in the post-monsoon (>80% of occurrences) and are dominated by days where winds were from the north-west (>75%). On these days (north-westerly winds during the post-monsoon) local (70%) transport emissions dominate (>75%). The large contribution of local transport emissions is likely driven by high $NO_x$ emissions in a shallow boundary layer, which acts to traps pollutants at the surface. While the increased non-local contribution of transport emissions (30%) is likely driven by trajectories (air masses) that travel towards Delhi, remaining close to the surface, accumulating increased emissions that are not quickly lost due to decreased sunlight and therefore less abundant OH during winter months.

## 4.3 Comparison to previous work

The results of this study are in line with previous work by Jethva *et al.* (2018) and Sembhi *et al.* (2020). Jethva *et al.,* (2018) used 3-day HYSPLIT back trajectories which were released from 3 different altitudes (100 m, 500 m and 1500 m) in Delhi each day between October-November 2013-2016 at 13:30 local time. Trajectories were grouped according to the 24-hour averaged $PM_{2.5}$ concentration at the US Embassy in Delhi (0 to <100 µg m$^{-3}$, 100 to <200 µg m$^{-3}$, 200 to <300 µg m$^{-3}$ and >300 µg m$^{-3}$). In most cases, near surface trajectories passed over crop burning regions in north-west India (Punjab and Haryana) (52%, 81%, 89% 84%, respectively). Thus, indicating air masses passing over crop burning regions are associated with increased $PM_{2.5}$ concentrations in Delhi. In addition, Jethva *et al.,* (2018) estimated that trajectories took around 14-22 hours to be advected from Punjab and Haryana to Delhi indicating the potential for the advection of $NO_x$ emissions to Delhi too. Sembhi *et al.* (2020) used a model to simulate air quality in Delhi during a poor AQ episode in 2016 with and without the implementation of the SSWA. They found that timing shift in agricultural burning in north-west India caused by the introduction of the SSWA contributed only around 3% to the poor AQ observed, indicating that this was largely driven by other factors. We also find that trajectories originating from the north-west during post-monsoon months have a polluted footprint in our analysis of satellite data and emissions. Both previous studies from Jethva *et al.* (2018) and Sembhi *et al.* (2020) suggest the potential for the advection of $NO_x$ fire emissions towards Delhi from source regions. However, within our work we do not see an impact from the advection of $NO_x$ fire emissions, which could be for several reasons. Firstly, Jethva *et al.* (2018) do not consider the interaction of boundary layer height and trajectory height when including trajectories in their analysis. Whereas, in this study, fire (and anthropogenic) emissions are only accumulated if the trajectory is within the boundary layer, which is very shallow during the post-monsoon. As a result, few trajectories are accumulated. Since fire emissions are buoyant and create plumes, which often extend above the boundary layer, the influence of fires may be underestimated in this study. Secondly, Sembhi *et al.* (2020) focussed on $PM_{2.5}$, which has a much longer atmospheric-lifetime than $NO_x$ (days to weeks compared with hours to days). In our results, the shorter atmospheric lifetime of $NO_x$, relative to $PM_{2.5}$, leads to a smaller contribution in the advection of $NO_x$ from fires, occurring in north-west India during the post-monsoon, towards Delhi.. Finally, and arguably most importantly, fire emissions are generated using polar orbiting satellites which have a single daytime overpass and thus may miss fires which have a short burn time; fire emissions inventories currently struggle to detect agricultural waste burning fires due to their small size and often short burn times (Zhang et al., 2020; Liu et al., 2020). Although we have used VIIRS in this study (which is able to detect smaller, lower temperature fires than MODIS) the total emissions from agricultural waste burning may still be underestimated (Zhang et al., 2020; Liu et al., 2020). In addition, inventories struggle with fire detection during hazy periods, particularly those which use active fire detection (such as FINNv2.5 used in this study) leading to underestimations in fire emissions. This is supported by the large range in fire emissions estimates for November 2018, ranging from 0.63 Tg to 5.52 Tg. To accurately quantify the influence of fire emissions on Delhi AQ in the post-monsoon fire emissions inventories need to overcome these known issues. However, with the introduction of geostationary satellites and sensors which can continuously detect smaller fires (e.g. Himawari) it should be easier to constrain the emissions from agricultural waste burning in the future.

## 4.4 Implications for Policy to Control Air Pollution over Delhi

The post-monsoon season is most polluted. During this time, trajectories arrive in Delhi from the north-west. Satellite $TCNO_2$ indicates this is likely due to a combination of high emissions from agricultural burning in Haryana and Pubjab alongside high local emissions in Delhi. Whereas $NO_x$ emissions datasets indicate that transport emissions dominate under all wind-directions and seasons, indicating that emissions reductions in this sector would lead to the largest benefits. To improve local AQ in Delhi both local and regional transport and agricultural waste burning emissions would need to be reduced.

## Code Availability

Code used to generate the figures in this paper is available on request.

## Data Availability

Data used in this paper is available on request.

## Author Contributions

AMG processed emissions datasets, adapted and ran back-trajectory code, previously written by RJP, and created all figures. RJP downloaded and processed satellite data. SD downloaded and processed ERA-Interim data. AMG wrote the manuscript with input from RJP, SD and MPC. MP provided TOMCAT 3D OH fields from model simulations which WF helped set-up. VS provided ground-based $NO_2$ observations from Delhi. YC and OW helped with $NO_2$ lifetime calculations. RS provided guidance and feedback on the work. GB provided SAFAR $NO_x$ emissions.

## Competing Interests

The authors declare that they have no conflict of interest.

## Acknowledgements

This work was supported by the UK Natural Environment Research Council (NERC) (NE/P016391/1 and NE/R001782/1) and by UK Research and Innovation through the Science and Technology Fund (ST/V00266X/1).

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

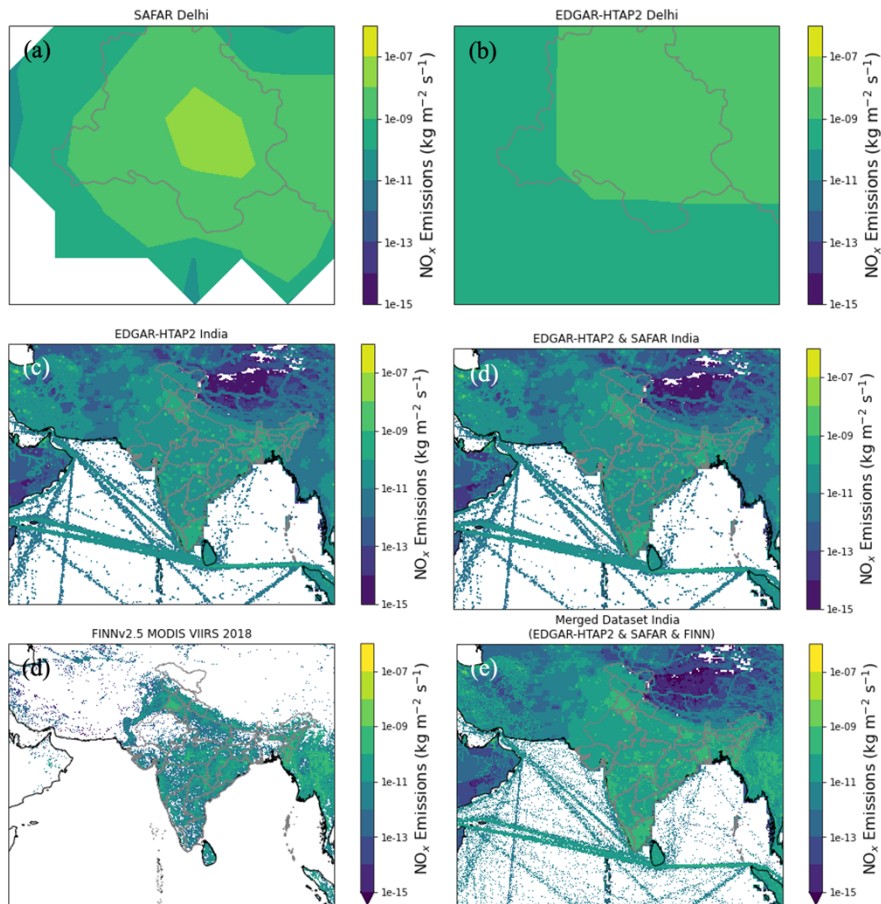

Figure 1. (a) NO$_x$ emissions (kg m$^{-2}$ s$^{-1}$) within Delhi from the System of Air Quality and Weather Forecasting And Research (SAFAR) (Beig, 2018) for 2018 regridded from 400 m to 10 km resolution. (b) NO$_x$ emissions within Delhi from the Emission Database for Global Atmospheric Research with Task Force on Hemispheric Transport of Air Pollution version 2.2 (EDGAR-HTAP2) (Janssens-Maenhout et al., 2015) emissions for 2010 at 10 km resolution. (c) Global NO$_x$ emissions from EDGAR-HTAP2 for 2010 at 10 km resolution. (d) Merged EDGAR-HTAP2 and SAFAR NO$_x$ emissions, where EDGAR-HTAP2 NO$_x$ emissions within Delhi have been replaced with NO$_x$ emissions from SAFAR. (e) Global fire emissions provided by the FINNv2.5 dataset for 2018 at 10 km resolution. (e) Merged anthropogenic and fire NO$_x$ emissions dataset, where SAFAR, EDGAR-HTAP2 and FINNv2.5 emissions have been combined.

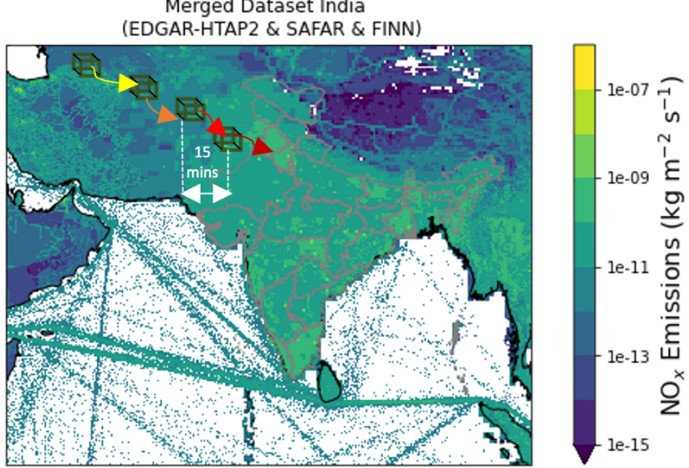

Figure 2. Back-trajectory method used in this study. Gridded emissions from a bottom-up inventory (Figure 1e) are summed every 15-minutes (15 mins) along the back-trajectory path (arrows) when the air parcel is within the boundary layer (indicated by boxes). NO$_x$ accumulates along the trajectory path towards Delhi (indicated by line shading). Satellite tropospheric column NO$_2$ (TCNO$_2$) is also summed in a similar way.

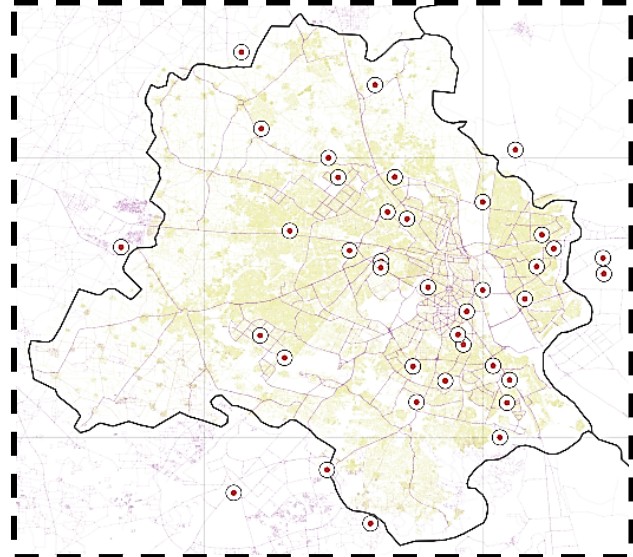

Figure 3. Map showing the location of NO₂ monitoring sites across the Delhi region (n_sites=36). In this study we use the daily 07:30-08:30 UTC (13:00-14:00 local time) median NO₂ concentration (calculated using all sites shown) to determine the timing of high pollution days in Delhi. We define high pollution days as days where the NO₂ concentration exceeds the 90th percentile of NO₂ concentrations between 2018 and 2019.

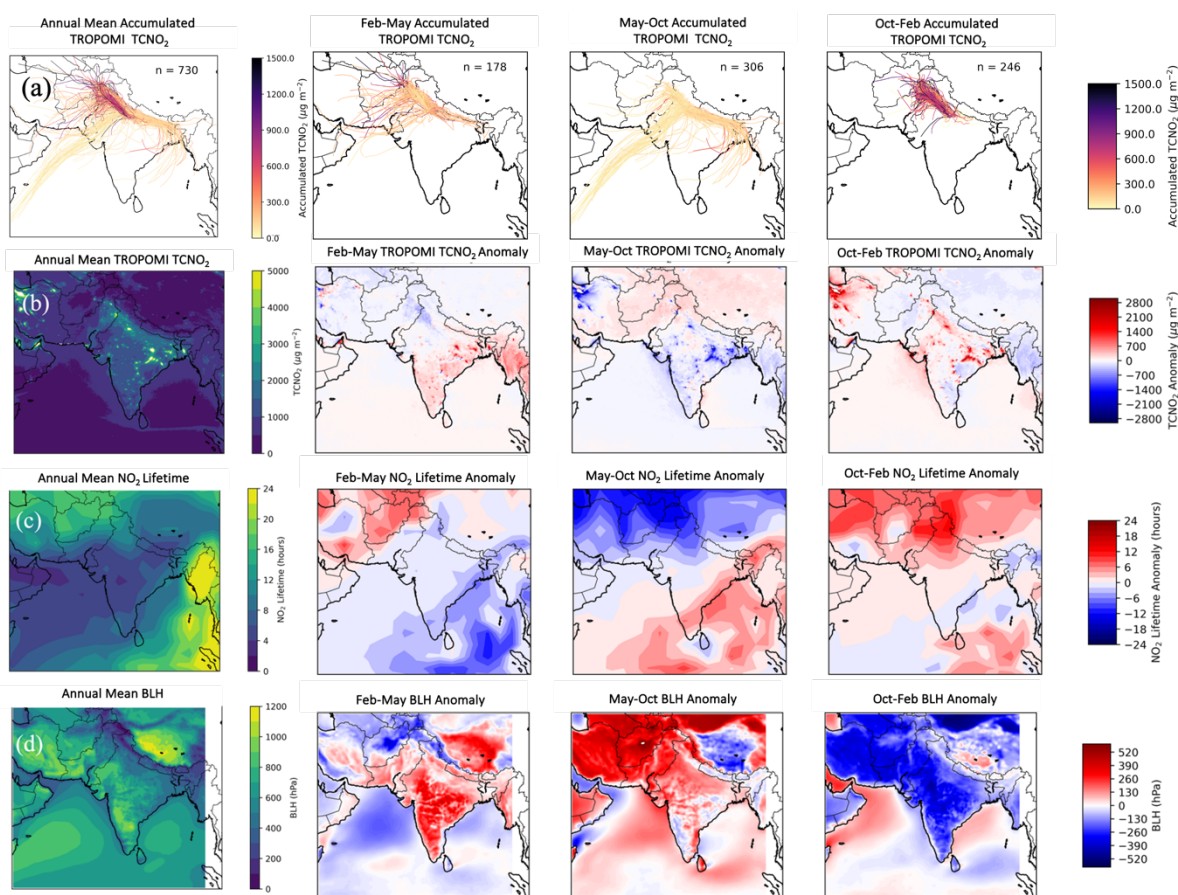

Figure 4. (a) Total annual (and seasonal) accumulated TROPOMI TCNO$_2$ (µg m$^{-2}$) arriving in Delhi along 4-day back trajectories. Trajectories are coloured by their total accumulated TCNO$_2$, with darker trajectories indicating higher levels of accumulated TCNO$_2$. (b) Mean annual (and seasonal) TCNO$_2$ (anomaly) (µg m$^{-2}$), (c) mean annual (and seasonal) NO$_2$ lifetime (anomaly) (hours) and (d) mean annual (and seasonal) boundary layer height (BLH) (anomaly) (hPa) compared to all seasons. All panels show the time period of 2017 and 2018 annually and seasonally - during the pre-monsoon (Feb-May), monsoon (May-Oct) and post-monsoon (Oct-Feb) seasons, respectively.

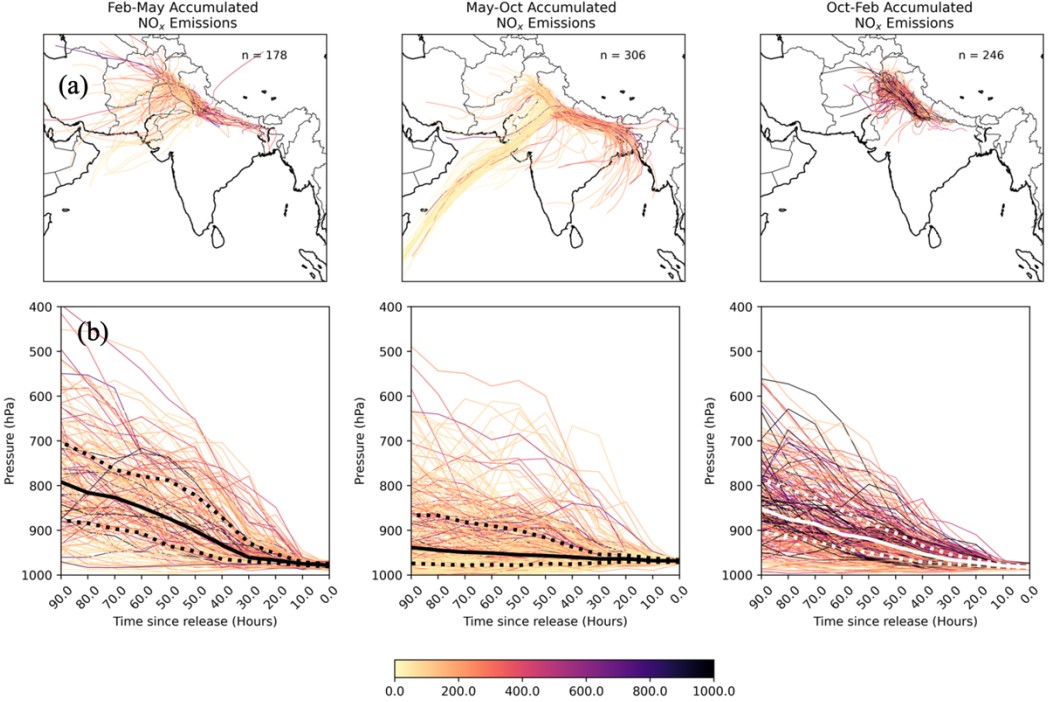

**Figure 5. (a)** Total accumulated NO$_x$ emissions (µg m$^{-2}$) arriving in Delhi in 2017 and 2018 and **(b)** pressure level of air parcels as they converge on Delhi, just above the surface (~1000 hPa) at hour 0 in pre-monsoon (Feb-May), monsoon (May-Oct) and post-monsoon (Oct-Feb) seasons. Median (black/white line), 25$^{th}$ and 75$^{th}$ percentiles (black/white dashed lines) of the trajectory pressure are also indicated. Trajectories are coloured by their total accumulated emissions, with darker trajectories indicating higher levels of accumulated NO$_x$.

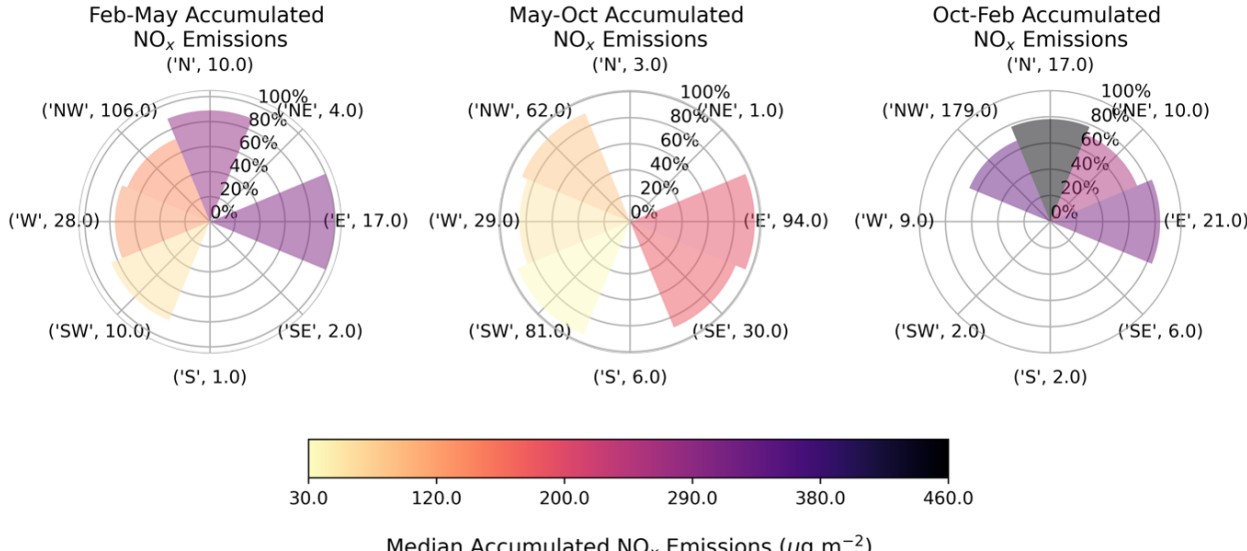

**Figure 6.** Wind rose of median accumulated NO$_x$ emissions (µg m$^{-2}$) from 4-day back trajectories with 6-hr time steps arriving at Delhi in 2017 and 2018 for pre-monsoon (Feb-May), monsoon (May-Oct) and post-monsoon (Oct-Feb) seasons. Total accumulated emissions are indicated by the shading of the segments. The area of the segment indicates the non-local contribution to the total integrated emissions. The percentage of local emissions is indicated on the circles. The number of days on which each wind direction occurs in each season is also indicated in brackets. For example, NW in the post-monsoon is very polluted (~350 µg m$^{-2}$), 65% of total integrated emissions are non-local and this wind direction occurs on 179 days.

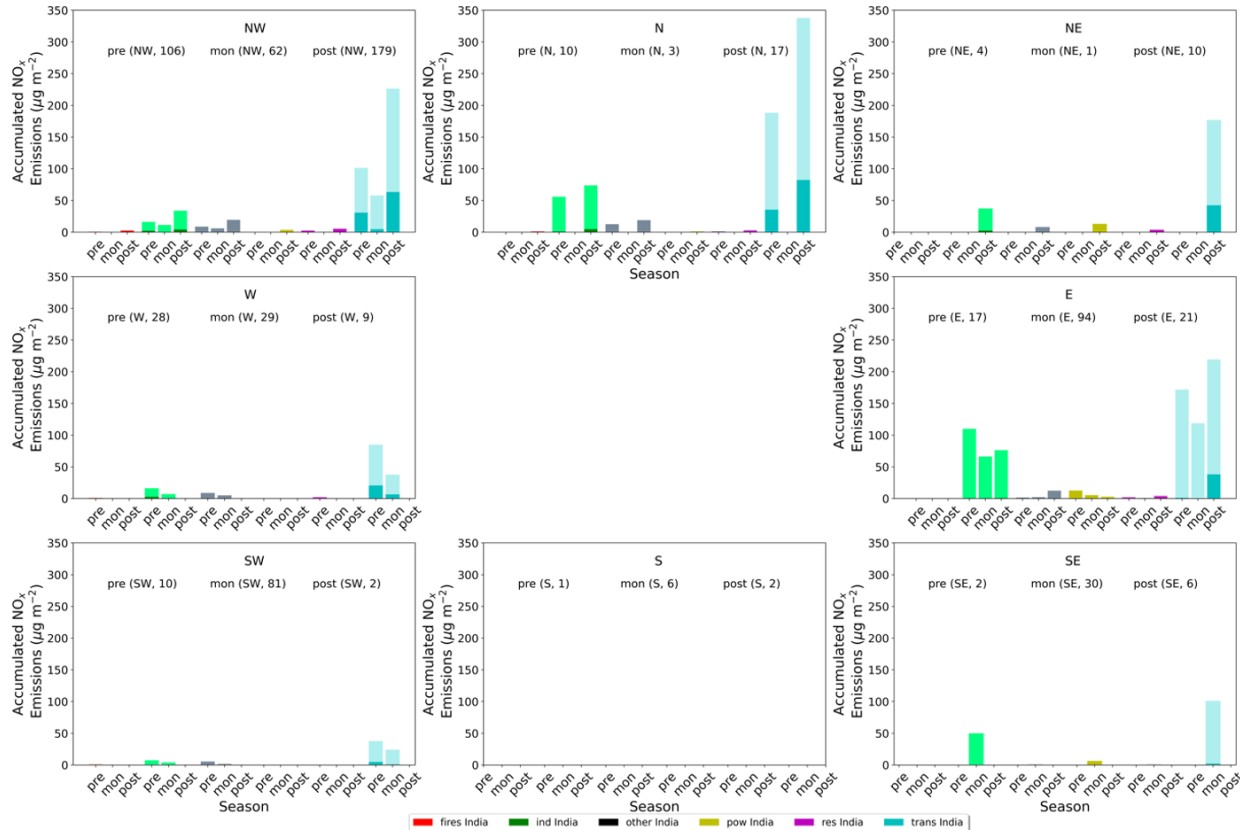

**Figure 7. Sector-specific accumulated NO<sub>x</sub> emissions (µg m⁻²) from 4-day back trajectories with 6-hr time steps arriving in Delhi in 2017 and 2018 in pre-monsoon (pre) (Feb-May), monsoon (mon) (May-Oct) and post-monsoon (post) (Oct-Feb). Light shading indicates accumulated NO<sub>x</sub> emissions originating within Delhi and darker shading indicates accumulated NO<sub>x</sub> emissions originating from outside of Delhi. Sectors included are fire (FIRES) (red), industrial (IND) (green), other (OTHER) (black), power generation (POW) (gold), residential (RES) (magenta) and transportation (TRANS) (cyan) emissions. The number of days for which each wind direction occurs in each season is indicated in brackets. For example, north-westerly winds occur on 179 days during the post-monsoon and transport emissions dominate (~80 µg m⁻²). The contribution from transport is split into ~55 µg m⁻² that are non-local (India) and ~25 µg m⁻² that are local (Delhi).**

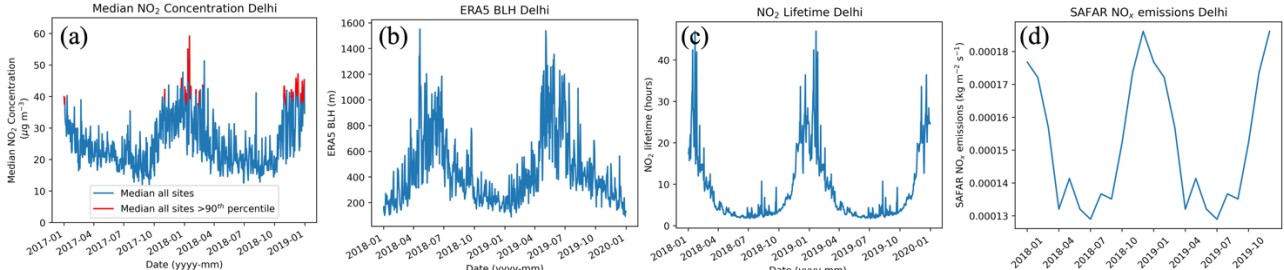

**Figure 8. (a) Daily median NO₂ concentrations (µg m⁻³) in Delhi (blue). Concentrations above the 90ᵗʰ percentile of observations are indicated in red. (b) Daily mean boundary layer height (BLH) (m) in Delhi from ERA5 re-analysis, (c) Daily mean NO<sub>x</sub> lifetime (hours) in Delhi and (d) Monthly mean NO<sub>x</sub> emissions (kg m⁻² s⁻¹) from the SAFAR emissions inventory. All panels show the time-period between 1ˢᵗ January 2017 and 31ˢᵗ December 2018.**

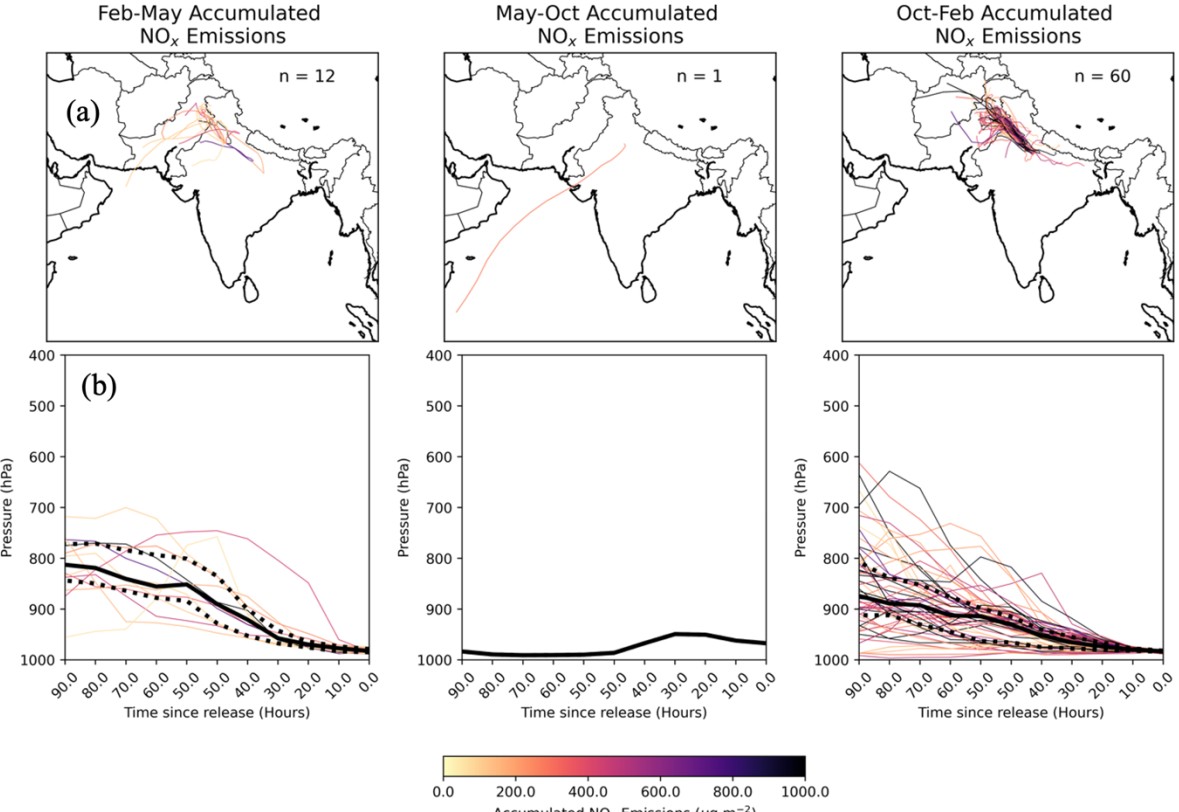

**Figure 9. (a) Total accumulated NO$_x$ emissions (µg m$^{-2}$) arriving at Delhi during high pollution events in in 2017 and 2018, (b) pressure level of air parcels as they converge on Delhi just above the surface (~1000 hPa) at hour 0 in pre-monsoon (Feb-May), monsoon (May-Oct) and post-monsoon (Oct-Feb). Median (black line), 25[th] and 75[th] percentiles (black dashed lines) of the trajectory pressure are also indicated. Trajectories are coloured by their total accumulated emissions, with darker trajectories indicating higher levels of accumulated NO$_x$.**

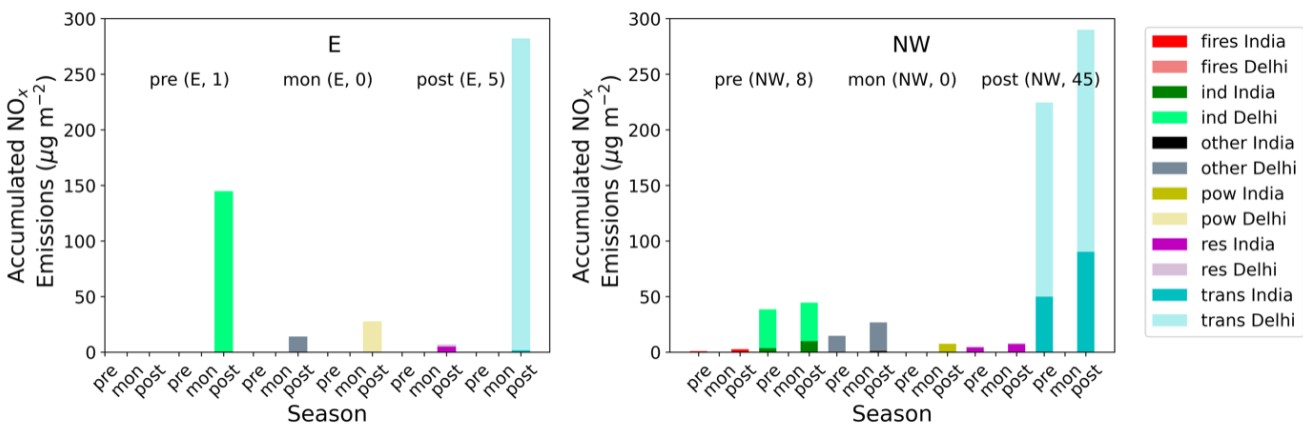

**Figure 10. Sector-specific accumulated NO$_x$ emissions (μg m$^{-2}$) from 4-day back trajectories with 6-hr time steps arriving at Delhi during high pollution events in 2017 and 2018 in the pre-monsoon (pre) (Feb-May), monsoon (mon) (May-Oct) and post-monsoon (post) (Oct-Feb). Light shading indicates accumulated NO$_x$ emissions originating within Delhi and darker shading accumulated NO$_x$ emissions from outside of Delhi. Sectors included are fire (FIRES) (red), industrial (IND) (green), other (OTHER) (black), power generation (POW) (gold), residential (RES) (magenta) and transportation (TRANS) (cyan) emissions. The number of days for which each wind direction occurs in each season is indicated in brackets. For example, north-westerly winds occur on 45 days during high pollution days in the post-monsoon and transport emissions dominate (>250 μg m$^{-2}$). The contribution from transport is split into ~80 μg m$^{-2}$ that are non-local (trans India) and ~200 μg m$^{-2}$ that are local (trans Delhi).**

**Tables**

**Table 1. Details of the anthropogenic and fire emissions datasets used in this study. These were combined to generate daily emissions for India (and the surrounding region).**

| Dataset | Year | Resolution (km) | Region | Dataset Info |
|---|---|---|---|---|
| **EDGAR-HTAP2** | 2010 | 10 | Global | Anthropogenic Emissions, Monthly |
| **SAFAR** | 2018 | 0.4 | Delhi | Anthropogenic Emissions, Monthly |
| **FINNv2.5 (MODIS VIIRS)** | 2018 | 10 | Global | Wildfire Emissions, Daily |

| Season | Dominant wind direction (% of days) | Key Source Sectors (% contribution) | Local/non-local contribution to accumulated emissions for Key Source Sector (% contribution) |
|---|---|---|---|
| Pre-monsoon | NW (54%) | Transport (78%) | Local (30%)/Non-local (70%) |
| Monsoon | E (31%) | Transport (61%) Industry (34%) | Local (>99%)/Non-local (<1%) Local (>99%)/Non-local (<1%) |
| | SW (27%) | Transport (81%) Industry (13%) | Local (96%)/Non-local (4%) Local (97%)/Non-local (3%) |
| | NW (20%) | Transport (76%) Industry (15%) | Local (8%)/Non-local (92%) Local (96%)/Non-local (4%) |
| Post-monsoon | NW (73%) | Transport (77%) Industry (12%) | Local (72%)/Non-Local (28%) Local (88%)/Non-Local (12%) |

| Season | Dominant wind direction (% of days) | Key Source Sectors (% contribution) | Local/non-local contribution to accumulated emissions for Key Source Sector (% contribution) |
|---|---|---|---|
| Pre-monsoon | NW (54%) | Transport (78%) | Local (30%)/Non-local (70%) |
| Monsoon | E (31%) | Transport (61%)<br>Industry (34%) | Local (>99%)/Non-local (<1%)<br>Local (>99%)/Non-local (<1%) |
| | SW (27%) | Transport (81%)<br>Industry (13%) | Local (96%)/Non-local (4%)<br>Local (97%)/Non-local (3%) |
| | NW (20%) | Transport (76%)<br>Industry (15%) | Local (8%)/Non-local (92%)<br>Local (96%)/Non-local (4%) |
| Post-monsoon | NW (73%) | Transport (77%)<br>Industry (12%) | Local (72%)/Non-Local (28%)<br>Local (88%)/Non-Local (12%) |

**Table 2. Summary of key wind-directions in the pre-monsoon (Feb-Apr), monsoon (May-Sept) and post-monsoon (Oct-Jan) monsoon (as % of total days in each season). The key source sectors with the largest contribution to the overall total accumulated emissions are also indicated (as % contribution) alongside the contribution of local (Delhi) and non-local (rest of India) accumulated emissions to overall accumulated emissions for that sector. For example, in the post-monsoon NW winds occur 73% of days, transport emissions contribute 77% of the overall integrated emissions and 72% of transport emissions are local, while 28% are non-local. Seasonal sector-specific accumulated $NO_x$ emissions ($\mu g\ m^{-2}$) are calculated from 4-day back trajectories with 6-hr time steps arriving in Delhi in 2017 and 2018 in pre-monsoon, monsoon and post-monsoon periods.**
