# Peer review of "Quantifying effects of long-range transport of NO2 over Delhi using back-trajectories and satellite data"

_EGUsphere, 2023_

## Author Comment (AC1)

*We would firstly like to thank the reviewer for their time and useful comments. We will address comments in blue and include any changes to the manuscript in green. We have combined the comments from both reviewers and respond to each below.*

*In addition, we found a minor bug in the code which has a minor effect on the results. We have updated the text and figures in line with these changes.*

**General Description:**

The authors use back-trajectories and satellite data to investigate the long-range transport of NO2 over Delhi. The study is comprehensive and findings from this study will help promote targeted mitigation measures. Overall, the paper is well written but would benefit from a detailed discussion on the novelty and some aspects of the methodology. So, I can only recommend this paper for publication after major revisions.

**General Comments:**

In the abstract, the authors mention that nitrogen oxides are hazardous air pollutants for health and form secondary aerosols and ozone. There is no mention of this in the introduction or discussion on the implications of the study on health and secondary pollutants.

We have removed this statement given the mixed evidence for a causal link between $NO_2$ exposure and health impacts. This means it's difficult to put the study's results into the context of a health impact.

The authors mention a few studies in the introduction which have an overlap with this study and so the novelty of this study is not clear. The authors should describe how this study adds value to the existing literature.

We have updated the manuscript to clarify this further.

Here our approach is similar to Stirling *et al*. (2020) and Graham *et al*. (2020). Our method can be split into 3 parts. For the first time we combine satellite tropospheric column $NO_2$ ($TCNO_2$) (top-down) with back trajectories (released from Delhi during 2017 and 2018). Secondly, we combine anthropogenic and fire emissions of $NO_x$ (bottom-up) with back-trajectories, in the same way as Stirling *et al*. (2020) and Graham *et al*. (2020). Both steps allow us to quantify the contribution of local and non-local emissions sources to poor $NO_x$ AQ in Delhi. And thirdly, we exclude sectors from the bottom-up anthropogenic emissions dataset in order to identify key source sectors contributing to poor $NO_x$ AQ in Delhi. This develops the methodology of Stirling *et al*. (2020) and Graham *et al*. (2020) further.  Section 2 describes the

datasets and method used, Section 3 presents our results and Section 4 summarises the implications of our findings.

There are two recently developed emission inventory datasets, one for road transport (Hakkim et al., 2021) and other for stubble burning (Kumar et al., 2021). The road transport dataset shows that existing emission inventories overestimate NOx emissions by a factor of 3 and this could potentially explain why the authors see large contributions from the transport sector. Could the authors do a comparison of the emission inventories or a sensitivity test to see how NOx emissions from this latest dataset would impact the study findings? Similarly, how does the stubble burning emission inventory compare to the fire emissions inventories. Could the role of agricultural fires be larger than what the authors have quantified?

We have compared the transport emissions burdens from the EDGAR-HTAP2 dataset with road transport emissions from Hakkim *et al.* (2021) [RTEII]. We find RTEII road transport NOx emissions are higher than EDGAR-HTAP2 transport NOx emissions burdens across most of India (Figure 1) and the total burdens are shown below:

| Dataset | Road Transport Emissions (Gg) | Transport Emissions (Gg) |
|---|---|---|
| RTEII | 1.10e3 | |
| EDGAR-HTAP2 | | 4.27e2 |

[Figure]

Figure 1. India Road Transport NOx emissions burdens (Gg) in Hakkim *et al.* (2021) [RTEII] and Transport NOx emissions in EDGAR-HTAP2. Finally, comparison of NOx emissions burdens (Gg) in RTEII and EDGAR-HTAP2.

[Figure]

Figure 2. Delhi region Road Transport NOx emissions burdens (Gg) in Hakkim et al., (2021) [RTEII], Transport NOx emissions in EDGAR-HTAP2 and Transport NOx emissions in SAFAR. Finally, comparison of NOx emissions burdens (Gg) in RTEII and SAFAR (RTEII-SAFAR) and EDGAR-HTAP2 and SAFAR (EDGAR-HTAP2-SAFAR).

For Delhi only, the NOx emissions burden from transport emissions in each dataset are as follows:

We also Delhi

| Dataset | Road Transport Emissions (Gg) | Transport Emissions (Gg) |
|---|---|---|
| SAFAR | | 21.5 |
| RTEII | 26.5 | |
| EDGAR-HTAP2 | | 18.2 |

plot the NOx

transport emissions burdens in each dataset (RTEII, EDGAR-HTAP2 and SAFAR) and compare the datasets we use to RTEII and then to themselves (RTEII-SAFAR and EDGAR-HTAP2-SAFAR). This indicates that the spatial pattern of emissions in SAFAR is very similar to RTEII, but higher in central Delhi and lower in the more suburban regions. EDGAR-HTAP2 on the other hand does not capture the spatial pattern of RTEII, with no clear NOx gradient in Delhi itself. The comparison plots indicate that RTEII NOx emissions are considerably lower than SAFAR in central Delhi (-1 to -2 Gg) but in suburban regions emissions are higher than in SAFAR (0 to 2 Gg). In agreement with this, EDGARHTAP2 is also considerably lower than SAFAR in central Delhi (-1 to -2 Gg) but higher in suburban regions (0 to 1 Gg).

We also compare Kumar *et al.* (2021) agricultural waste burning emissions for October-November 2017 with FINNv2.5 MODIS VIIRS daily fire emissions for October-November 2018 (the dataset we used in this study) (Figure 3) and FINNv2.5 MODIS (Figure 4).

[Figure]

Figure 3. October-November 2017 agricultural waste burning NOx emissions burdens (Gg) in Kumar *et al.* (2021) and October-November 2018 FINNv2.5 MODIS VIIRS NOx fire emissions (Gg). Finally, comparison of October-November NOx emissions burdens (Gg) in FINNv2.5 MODIS VIIRS 2018 and Kumar *et al.* (2021).

[Figure]

[Figure]

Figure 4. October-November 2017 agricultural waste burning NOx emissions burdens (Gg) in Kumar *et al.* (2021) and October-November 2018 FINNv2.5 MODIS NOx fire emissions (Gg). Finally, comparison of October-November NOx emissions burdens (Gg) in FINNv2.5 MODIS 2018 and Kumar *et al.* (2021).

We find that the total NOx fire emissions burden from the dataset we have used (FINNv2.5 MODIS VIIRS) is substantially higher (177.20 Gg) than Kumar *et al.* (2021) (37.27 Gg). The highest emissions in FINNv2.5 MODIS VIIRS are in south-west Punjab, whereas emissions are more evenly distributed across Punjab in Kumar et al. (2021). We also compare FINNv2.5 MODIS (without VIIRS) to Kumar *et al.* (2021), since VIIRS can detect very small fires. The emissions burden in FINNv2.5 MODIS is considerably lower than in FINNv2.5 MODIS VIIRS across most of Punjab (MODIS: 109.99 Gg, MODIS VIIRS: 177.20 Gg), however the total emissions burden is still considerably higher than Kumar *et al.* (2021) (37.27 Gg).

There are minor grammatical errors in the manuscript, and I have pointed out some below.

**Specific Comments:**

Line 93. Just to confirm, it is 'daily' and not 'hourly' PM2.5 peaking at over 1000 ug/m3?

This is correct, we have updated the manuscript accordingly.

Winter air pollutant concentrations are particularly high, with daily mean particulate matter with a diameter less than 2.5 micrometres ($PM_{2.5}$) concentrations of 100-200 $\mu g\ m^{-3}$ (Singh et al., 2021), and hourly concentrations peaking at over 1000 $\mu g\ m^{-3}$ (Bikkina et al., 2019).

Lines 96-99. The authors should consider including additional literature to support the use of long-term satellite obseravations and focused on NO2 in Delhi (For example, Vohra et al., 2021).

We have added the extra supporting literature you suggested.

The high pollution levels in Delhi are also apparent in long-term satellite observations of tropospheric column $NO_2$ ($TCNO_2$) (Vohra et al., 2021).

Line 114. Stirling et al. (2020) and Graham et al. (2020) 'extended this methodology'. This is not very clearly written and sounds like the only addition is that it now includes PM2.5 in addition to NO2.

We have updated this sentence to clarify the changes that were made to the methodology.

Subsequently Stirling *et al.* (2020) and Graham *et al.* (2020) developed this methodology further, including the accumulation of emissions along the trajectory path by combining back trajectories with bottom-up emissions inventories, to analyse changes in $NO_x$ and $PM_{2.5}$ concentrations.

Lines 119-127. Referring to the above comment, it is not exactly clear what the two studies do and so it is difficult to understand how the authors' approach is similar and/or different. The authors should also refer to Bikkina et al. (2019) here or in lines 75-85 to explain clearly what has been done before and how does this study contributes to new knowledge.

We have clarified the similarities and differences of this study to Stirling *et al.* (2020) and Graham *et al.* (2020).

Here our approach is similar to Stirling *et al.* (2020) and Graham *et al.* (2020). Our method can be split into 3 parts. For the first time we combine satellite tropospheric column $NO_2$ ($TCNO_2$) (top-down) with back trajectories (released from Delhi during 2017 and 2018). Secondly, we combine anthropogenic and fire emissions of $NO_x$ (bottom-up) with back-trajectories, in the same way as Stirling *et al.* (2020) and Graham *et al.* (2020). Both steps allow us to quantify the contribution of local and non-local emissions sources to poor $NO_x$ AQ in Delhi. And thirdly, we exclude sectors from the bottom-up anthropogenic emissions dataset in order to identify key source sectors contributing to poor $NO_x$ AQ in Delhi. This develops the methodology of Stirling *et al.* (2020) and Graham *et al.* (2020) further.

Lines 126-127. Confusing. Please rephrase.

Section edited based on previous comment.

Here our approach is similar to Stirling et al. (2020) and Graham et al. (2020). Our method can be split into 3 parts. For the first time we combine satellite tropospheric column NO2 (TCNO2) (top-down) with back trajectories (released from Delhi during 2017 and 2018). Secondly, we combine anthropogenic and fire emissions of NOx (bottom-up) with back-trajectories, in the same way as Stirling et al. (2020) and Graham et al. (2020). Both steps

allow us to quantify the contribution of local and non-local emissions sources to poor NOx AQ in Delhi. And thirdly, we exclude sectors from the bottom-up anthropogenic emissions dataset in order to identify key source sectors contributing to poor NOx AQ in Delhi. This develops the methodology of Stirling et al. (2020) and Graham et al. (2020) further.  Section 2 describes the datasets and method used, Section 3 presents our results and Section 4 summarises the implications of our findings.

Section 2.1 on anthropogenic NOx emissions is quite difficult to follow. This should be restructured to have an introductory statement "We created a merged emissions data … (Lines 134-135) and then have sections on Delhi emissions and global emissions. If the authors want to have fire emissions as a separate section, then this should be made clear in the sections above, something like Delhi (except fire) emissions.

We have restructured this section and added in a table that gives and overview of the dataset generated.

We created a merged emissions dataset for India containing both anthropogenic and daily fire emissions (Figure 1f). Table X summarises the datasets used to generate this dataset.

| Dataset | Year | Resolution (km) | Region | Dataset Info |
| --- | --- | --- | --- | --- |
| SAFAR | 2018 | 0.4 | Delhi | Anthropogenic Emissions, Monthly |
| EDGAR-HTAP2 | 2010 | 10 | Global | Anthropogenic Emissions, Monthly |
| FINNv2.5 (MODIS VIIRS) | 2018 | 10 | Global | Wildfire Emissions, Daily |

The authors do not mention natural sources such as lightning. Is their contribution negligible?

We only consider sources where emissions can be reduced using policies. Therefore we do not include natural sources of $NO_2$ or $NO_x$ like lightning or soil since these cannot be targeted for emissions reductions.

Section 2.1 should have a final section 2.1.4 on Merging of emissions to discuss the approach of combining these datasets. Currently, it is not clear what approach has been followed and what has been done to account for discontinuity in merging Delhi and global emissions datasets. The authors have included figures, but a quantitative assessment is also warranted.

We have updated the manuscript to clarify how merging was performed.

Monthly anthropogenic $NO_x$ emissions for Delhi are taken from the 2018 System of Air Quality and Weather Forecasting And Research (SAFAR) dataset (Beig, 2018) (Figure 1a).

Data for the SAFAR emission inventory were collected during a 37,500-hour campaign and are provided at very high (400 m) resolution over a region covering 70 km × 65 km. Emissions are provided for 26 different source sectors. Emissions are very detailed, for example emissions for transport were calculated using traffic volume collected by click counters across the region. Collected data was then converted into emissions using a Geographical Information System (GIS)-based statistical model. Since SAFAR provides a much more detailed, and up-to-date, inventory of emissions in Delhi we regrid SAFAR emissions to 10 km and then replace all EDGAR-HTAP2 emissions (see 2.1.2 for more details) in Delhi with SAFAR emissions using a simple mask method. We create an empty 10 km global grid and first add SAFAR emissions. Then we add EDGAR-HTAP2 emissions where the grid is still empty (i.e. where no SAFAR emissions were added). The combined dataset is resampled to daily resolution by interpolating monthly values to daily temporal resolution.

Line 134. The authors say daily fire emissions but what is the temporal resolution of the other anthropogenic emissions.

Both datasets are monthly and we resample them to daily by repeating monthly values, we have updated the manuscript.

Monthly anthropogenic $NO_x$ emissions for Delhi are taken from the 2018 System of Air Quality and Weather Forecasting And Research (SAFAR) dataset… We create an empty 10 km global grid and first add SAFAR emissions. Then we add EDGAR-HTAP2 emissions where the grid is still empty (i.e. where no SAFAR emissions were added). The combined dataset is resampled to daily resolution by interpolating monthly values to daily temporal resolution.

Section 2.2 Can the authors briefly describe the approach of Pope et al. (2018) to derive the 0.05-degree dataset? Is it oversampling or error-weighted tessellation? Any quality flags used? Which months in 2018/2019 are selected? The resolution of TROPOMI changed around mid-2019 and this has not been stated. How do the authors account for that?

Commented [AG1]: Ask Richard about this

The approach by Pope et al., (2018) uses an oversampling approach where individual satellite pixels are split up into sub-pixels and mapped onto a higher resolution regular grid than just using the pixel centre information alone. As such, the change in TROPOMI resolution has limited impact on the approach, as the satellite pixels is sliced into a similar number of sub-pixels. As the resolution is higher after mid-2019, then more pixels will be used in the oversampling compared to before mid-2019. However, they will cover the same spatial area just with more satellite pixels form mid-2019 onwards. Given the resolution of 7 km x 3.5 km and 5.5 km x 3.5 km is comparable to the grid resolution of 0.05° x 0.05° (i.e. approximately 5 km x 5km), it will have limited impact on the level-3 product generated from this method of Pope et al., (2018). In terms of data used and QA flags, data for all days is used (i.e. daily data). This is typically 14 or 15 orbits per day. The QA flags used were QA > 75 and cloud fraction <0.2. To make these points clearer in the manuscript, on Page 5 Line 200, we have added:

"The approach of Pope *et al.* (2018) uses an oversampling methodology where TROPOMI pixels are sliced into sub-pixels and mapped onto a high-resolution level-3 grid. Individual retrievals are filtered for a geometric cloud fraction < 0.2 and a quality control flag > 75 using all the available daily TROPOMI data for both years."

Line 202. The authors could consider using 'and' instead of either-or as both datasets are being used.

We have updated the manuscript to remove the use of 'either' however we feel the use of 'and' may be confusing to the reader as we combine the back trajectories with the top-down dataset separately to combining the back trajectories with the bottom-up dataset.

Lines 230-240. There are a few inconsistencies with the equation and description of the variables. Alpha should be alpha (subscript i). Variables such as N and E0 are described but are not in the equation. Also, not clear how E subscript N is linked to E subscript i.

We have updated the equation

The along-trajectory emission accumulation can be represented by Equation (1):

$$E = \sum_{i=1}^{N} [E_{i-1} + \emptyset_i . \Delta t . \alpha_i] e^{-\Delta t / \tau_i} \qquad (1)$$

where N (=384) and $E_0$ =0.0

$E_i$ is accumulated $NO_x$ (kg) at any given point *i* along the trajectory (with E at point 0 [$E_0$] being equal to 0), $\phi_i$ is the emissions flux of $NO_x$ (kg m$^{-2}$ s$^{-1}$) at point *i*, $\Delta t$ is the 15-minute time step, $\alpha$ at point *i* is the surface area of the grid box (m$^2$) and $\tau$ at point *i* is the specified $NO_x$ lifetime ($\tau$). Therefore, $E$ is total accumulated $NO_x$ mass (kg) and $N$ is the number of 15-minute time steps within the 4-day trajectory (384).

Line 265. The authors can be more specific. Is it the 8 wind directions seen in Figures 6-7?

We have clarified this.

Finally, the daily (06:00 UTC) total accumulated emission and *EDelhi/E* ratios from all sites were binned by 8 wind directions (north through to north-westerly) based on their end point in relation to Delhi.

Line 271. Figure 3 shows more than 36 sites and some of them are outside Delhi too. Can the authors resolve this discrepancy?

The dataset was provided to us as part of a previous work package on the PROMOTE project by Singh which used all 36 sites. We have updated the manuscript to clarify the sites are across the Delhi region not just Delhi itself.

Hourly averaged surface $NO_2$ concentration from 36 sites across the Delhi region for 2018 and 2019 (Figure 3) were obtained from the CAAQMS (Continuous Ambient Air Quality Monitoring Stations) portal (https://app.cpcbccr.com/ccr/#/caaqm-dashboard-all/caaqm-landing) of the Central Pollution Control Board (CPCB) of India.

Line 277. Is this local solar time or the local time? Should the authors consider a longer window (2, 3 or 4 hours) to ensure it includes the satellite overpass time?

This is local time, we clarify this in the statement regarding TROPOMI's overpass time. We take the mean for the 2 hours surrounding the overpass time so that we capture the nearest time steps to the satellite passing overhead.

As TROPOMI has a local overpass time of approximately 13:30, the average $NO_2$ between 13:00-14:00 hours was calculated for each site to represent the daily $NO_2$ corresponding to the overpass of TROPOMI.

Line 287. It is not exactly clear in the first instance where the BLH data is from. After reading the full paper, I know it is from ERA-Interim reanalysis dataset.

We have clarified this in the manuscript (line 277)

NOx emissions were only accumulated only if the trajectory path was within the boundary layer (which we determine based on ERA-Interim reanalysis).

Lines 290-300. This section would benefit cofrom more context for the range of values provided. Comparison to literature, perhaps?

Commented [AG2]: Compare to other papers on tcno2

We have added a comparison to European values.

Smaller urban $NO_2$ hotspots range between 3000 and 4000 μg m$^{-2}$, higher than values across large European hotspots which peak at 3500 μg m-2 (Pope et al., 2019).

Line 304. Do the authors mean 'east Indian coastline'?

We have updated this.

Along the east Indian coastline

Lines 319-325. I get the reasons for post-monsoon season but not for pre-monsoon. Can the authors explicitly discuss how does pre-monsoon season conditions degrade NO2? Is this of similar magnitude to post-monsoon or just worse compared to monsoon conditions?

The same mechanisms lead to degradation of NO2 AQ in the pre-monsoon as the post-monsoon, the magnitude is just smaller. We have updated the manuscript to clarify this.

During the pre-monsoon and, especially, post-monsoon seasons conditions are favourable for the degradation of $NO_2$ air quality. In the pre- and post-monsoon primary $NO_x$ emissions (e.g. from increased domestic heating, power demands and agriculture burning) are typically larger, there is a longer $NO_2$ lifetime (i.e. less chemical loss) and a shallower boundary layer, trapping emissions over Delhi and the wider IGP. This effect is largest in the post-monsoon, though it is also apparent in the pre-monsoon. In contrast, during the monsoon there are lower primary $NO_x$ emissions, a shorter $NO_2$ lifetime and increased atmospheric ventilation of the boundary layer, all of which yield lower pollution levels during the monsoon season.

Line 344. Do the authors mean that the 'results are in close agreement to those obtained with TCNO2 datasets'?

We have expanded our explanation.

We repeated this approach using the bottom-up emissions inventory (Figure 5) finding the results are generally in close agreement to the $TCNO_2$ results, with the highest accumulated emissions being observed in the pre- and post-monsoon.

Lines 428-445. Are the surface observations used to identify the high pollution days from the satellite overpass time window? Are these representative of the daily mean NO2 in Delhi? Are the BLH also for the same time window?

Yes, the median NO2 of all Delhi region sites calculated from the daily site mean between 13:00-14:00 LT (n_sites=36) is used to identify high pollution days. Though the 13:00-14:00 LT mean NO2 may not capture the true daily mean values it will indicate high pollution days in comparison to others (i.e. variability is still captured). Yes, the BLH is the 13:00-14:00 UTC mean. This will reflect a time of day when the BLH is well developed.

Lines 450-461. The authors should mention high "NOx" pollution days to make it clear that poor AQ and high pollution are with reference to NO2.

We have added this in.

The back trajectory emissions analysis indicates that high $NO_2$ pollution days are associated with trajectories from the north-west of Delhi (Figure 9a) in both the pre- (n=12 days) and post-monsoon (n=60 days). Note that we only include wind-directions with a sample size greater than or equal to 5 (easterly and north-westerly). Trajectories gradually descend towards the surface between 90 and 20 hours out from Delhi, remaining close to the surface (>950 hPa) until they reach Delhi (Figure 9b). However, during the post-monsoon trajectories are much closer to the surface from hour 90 (910-810 hPa) compared with the pre-monsoon (840-770 hPa), likely allowing increased accumulation of emissions. During post-monsoon high-pollution days, local (~70% local) transport emissions dominate (>75%) the total accumulated emissions (Figure 10). This is likely due to the trajectories remaining close to the surface for the final 24 hours, within a shallow boundary layer, at a period when the $NO_x$

atmospheric lifetime and emissions are increased. The contribution of other sectors to the remaining accumulated emissions (residential, industrial, other and power) is smaller but local emissions remain important (25-100%). There is also a negligible contribution (<2 g m$^{-3}$) from non-local (100%) fires under north-westerly winds. Overall, this suggests that non-local NO$_x$ emissions from long-range transport (advection) contribute to poor NO$_2$ AQ in Delhi but the accumulation of local emissions under a shallow boundary layer dominate.

Figure 3 caption should mention the number of sites in Delhi and why do the authors include sites outside of Delhi?

We include sites outside of Delhi as we were provided with a dataset which only gives the median NO2 concentration across those sites. As a result we are unable to update this to only include those within Delhi, though we wouldn't expect this to have a large impact on our results. We have added the following:

Map showing the location of NO$_2$ monitoring sites across the Delhi region (n_sites=36).

Figure 4. The top panel axis label should read NOx emissions. TROPOMI data was not used for 2017 but is mentioned in the caption.

We apologise and thank you for noticing our mistake, The panel should read TCNO2 – it is the satellite TCNO2 data. We have updated the panel accordingly.

[Figure]

Figures 7/10. Should the emissions currently shown as "India" be "Rest of India" or do they include "Delhi"?

The figure shows Delhi only against all of India.

All figure captions and text should read as "Feb-Apr" instead of "Feb_Mar_Apr".

We have updated all figure captions accordingly – see above for example

*We would firstly like to thank the reviewer for their time and useful comments. We will address comments in blue and include any changes to the manuscript in green.*

*In addition, we found a minor bug in the code which has a minor effect on the results. We have updated the text and figures in line with these changes.*

**Dear Editor,**

The submitted manuscript attempts to quantify the effects of long-range transport of NO2 over the Delhi region in northern India using satellite (TROPOMI) data and back-trajectory analysis (ROTRAJ). Authors use bottom-up NOx inventories, TROPOMI high-res data on tropospheric NO2, and back-trajectories for quantifying local and non-local sources of air pollution impacting Delhi, with a focus on different emission sectors. Authors find that the accumulated emissions are highest under westerly, north-westerly, and northerly flow during pre- (February - March) and post-

(October - January) monsoon periods. During the post-monsoon period, non-local (60%) emissions from the transport sector are found to be the largest contributor to the total accumulated emissions as high emissions, coupled with a relatively long NOx atmospheric lifetime and shallower boundary-layer favors the build-up of emissions along the trajectory path. The high-pollution episodes are also found to occur predominantly in the post-monsoon and more than 75% of these events are primarily caused by non-local sources.

One major inconsistency noticed in the paper is that paper starts (in the Introduction section) with a description of the post-monsoon agricultural burning in NW India and related aspects (without discussing the transport sector with associated statistics and known emissions). Towards the end of the paper, the authors find that the non-local transport sector dominates in NOx concentration in Delhi. Figure 4 TCNO2 anomaly plot for the post-monsoon/winter seasons showed positive anomalies along the transport pathways from agricultural burning areas in NW to Delhi. If the transport sector emission remains constant, more or less, around the year, the additional burden of NO2 may be attributed to the burning activities in the region. However, it is not captured in the sectorial analysis, indicating that the NO2 inventories in the model need to be updated. The authors should discuss the strengths and weaknesses of the models and inventories employed in the study. The results section should be structured into sub-sections according to the type of analysis and presented results.

The manuscript has improved in terms of the presentation, language, and results from the first version reviewed for a quick response.

The detailed comments are attached to this report. Authors are encouraged to consider them while revising the paper. The paper doesn't seem to require substantially major revision if the authors are confident and accurate in their methodology and derived results but need to strike a balance in discussing the transport sector vs. agricultural burning.

Thanks for the review opportunity.

**Detailed comments:**

Abstract, line 37: The wheat residue burning takes place in the northwestern states of Punjab and Haryana during pre-monsoon (April-May). Although not as great in numbers compared to the post-monsoon paddy burning, it is possible that some transport of NO2 from these fires may contribute to NO2 levels in Delhi.

Line 43-44: Do authors want to say that the substantial import of NO2 to Delhi is dominated by the transport sector outside the Delhi region? Or paddy burning? Or local transport within Delhi?

We have updated this to look at both datasets individually.

During the post-monsoon period the highest accumulated satellite TCNO$_2$ trajectories are advected from Punjab and Haryana, where satellite TCNO$_2$ is elevated, indicating the potential for the long-range transport of agricultural burning emissions to Delhi. However accumulated NO$_x$ emissions indicate local (70%) emissions from the transport sector are the largest contributor to the total accumulated emissions. High local emissions, coupled with a relatively long NO$_x$ atmospheric lifetime and shallow boundary-layer aid the build-up of emissions locally, and along the trajectory path. This indicates the possibility that fire emissions datasets may not capture emissions from agricultural wate burning in the north-west sufficiently to accurately quantify their influence on Delhi AQ.

Line 47-49: The vehicular transport activities are more or less similar during the entire year (of course, the number of vehicles has increased in the Delhi region year over year). The question is why during post-monsoon, the NOX levels shoot up. Does it mean that a shallow boundary layer and longer NOX lifetime increased the levels and its exposure to the populations? In the earlier statement, the author states that "surface daily NO2 observations indicate that high pollution episodes (> 90th percentile) occur predominantly in the post-monsoon and more than 75% of high pollution events are primarily caused by non-local sources." What are these non-local sources? Transport? Agricultural burning? What are their relative contributions (in %)?

We discuss the source in the following sentences and have added in relative contributions:

Analysis of daily ground-based NO2 observations indicates that high pollution episodes (> 90th percentile) occur predominantly in the post-monsoon and more than 75% of high pollution events are primarily caused by local sources. But there is also a considerable influence from non-local (30%) emissions from the transport sector during these periods.  Overall, we find that in the post-monsoon period, there is substantial accumulation of high local NOx emissions from the transport sector (70% of total emissions, 70% local), alongside the import of NOx pollution into Delhi (30% non-local).

Introduction:

Line 60: Rice stubble is known to contain silica, making it not suitable for consumption by animals, hence less valued. A shorter time window for switching from rice to wheat (enforced by the Punjab Preservation of subsoil water act), not much value in rice stubble, quick turnaround, and not much labor, are some of the prime factors for farmers to resort to burning during post-monsoon.

We have updated the manuscript accordingly:

However, rice stubble is generally burned to clear the land quickly as the high silica content of rice stubble means it's not suitable for animal consumption and burning is both more economically viable and quicker than manual removal (Ahmed et al., 2015).

Line 63: these numbers were based on the thermal anomaly/fire product of MODIS.

Line 73: Make sure (throughout the manuscript) the order of the citations follows the journal standards, i.e., increasing order by the year of publication.

We have updated citations in line with this.

Line 80: This (shallower boundary layer) is largely true for the entire IGP (and not just for Delhi).

We have updated this:

Additionally, a shallow boundary layer over Delhi and the surrounding IGP region during autumn and winter months, low wind-speeds (1-3 ms$^{-1}$) and poor ventilation and mixing means air pollutants from the IGP advected towards Delhi remain close to the surface, aiding the build-up of pollutants.

Section 2.1.1 & 2.1.2: At this point, it is not clear nor described how the merging of high-res SAFAR inventories are merged with the EDGAR-HTAP2 dataset.

We have updated this section to explain the method used.

Since SAFAR provides a much more detailed, and up-to-date, inventory of emissions in Delhi we regrid SAFAR emissions to 10 km and then replace all EDGAR-HTAP2 emissions (see 2.1.2 for more details) in Delhi with SAFAR emissions using a simple mask method. We create an empty 10 km global grid and first add SAFAR emissions. Then we add EDGAR-HTAP2 emissions where the grid is still empty (i.e. where no SAFAR emissions were added). The combined dataset is resampled to daily resolution by interpolating monthly values to daily temporal resolution.

Section 2.2: TROPOMI spatial resolution was upgraded to 5.5 x 3.5 km. Please check with its ATBD/documentation.

See response to reviewer 1 comment (section2.2).

Section 3:

Line 291: Can authors provide the TCNO2 units in Dobson Unit too?

We convert the provided units (moles/m-2) to ug/m-2 for the analysis. Most studies present their results in moles/m2 therefore we have compared the values we observe over India to work by Pope et al., (2019) to provide some context.

Line 291:299: Can authors attribute here the seasonal anomalies in TCNO2 to changes in emission and meteorology? Since this is the total tropospheric column quantity, BLH will have no effect (like in surface PM2.5 or trace gases) in the columnar measurements.

We believe this is not the case. BLH will have an impact on the columnar measurements as the satellite is vertically sensitive so changes in BLH could trap

NO2 close to the surface where the satellite is more/less sensitive – therefore changing the columnar measurement due to BLH changes. In addition, changes in BLH could cause venting of NO2 aloft which would also affect the satellite measurements.

Line 319-323: Agricultural burning in northwestern India is also a major source author is missing mentioning here.

We have added this suggestion in.

In the pre- and post-monsoon primary $NO_x$ emissions (e.g. from increased domestic heating, power demands and agriculture burning) are typically larger…

Figure 5: Similar back-trajectory analysis was also shown by Jethva et al. (2018) AAQR paper for the post-monsoon season, finding that airmass arriving at Delhi at different altitudes were advected from northwest intercepting the agricultural burning regions of Punjab and Haryana. Also, the BT analysis revealed that it takes ~15-16 hours to less than a day for the airmass over Punjab to arrive at Delhi. Given the similar lifetime of NOX, the enhancement in NOX levels felt over Delhi may be likely attributed to agricultural burning. Plus, the results of descending air masses along the transport pathways are also consistent between the two studies. These all make sense. I believe it is worth mentioning here.

We have included the results of Jethva et al., 2018 in our discussion:

The results of this study are in line with previous work by Jethva et al. (2018) and Sembhi et al. (2020). Jethva et al., (2018) used 3-day HYSPLIT back trajectories which were released from 3 different altitudes (100 m, 500 m and 1500 m) in Delhi each day between October-November 2013-2016 at 13:30 local time. Trajectories were grouped according to the 24-hour averaged PM2.5 concentration at the US Embassy in Delhi (0 to <100 g m-3, 100 to <200 g m-3, 200 to <300 g m-3 and >300 g m-3). In most cases, near surface trajectories passed over crop burning regions in north-west India (Punjab and Haryana) (52%, 81%, 89% 84%, respectively). Thus, indicating air masses passing over crop burning regions are associated with increased PM2.5 concentrations in Delhi. In addition, Jethva et al., (2018) estimated that trajectories took around 14-22 hours to be advected from Punjab and Haryana to Delhi indicating the potential for the advection of NOx emissions to Delhi too. Sembhi et al. (2020) used a model to simulate air quality in Delhi during a poor AQ episode in 2016 with and without the implementation of the SSWA. They found that timing shift in agricultural burning in north-west India caused by the introduction of the SSWA contributed only around 3% to the poor AQ observed, indicating that this was largely driven by other factors. We also find that trajectories originating from the north-west during post-monsoon months have a polluted footprint in both the satellite and emissions analysis. Both previous studies suggest the potential for the advection of NOx fire emissions towards Delhi from source regions. However, within our work we do not see an impact from the advection of NOx fire emissions, which could be for several reasons. Firstly, Jethva et al. (2018) do not consider the interaction of boundary layer height and trajectory height when including trajectories in their analysis. Whereas, in this study, fire (and anthropogenic) emissions are only accumulated if the trajectory is within the boundary layer, which is very shallow during the post-monsoon. As a result, few trajectories are accumulated. Since fire emissions are buoyant and create plumes, which often extend above

the boundary layer, the influence of fires may be underestimated in this study. Secondly, Sembhi et al. (2020) focussed on PM2.5, which has a much longer atmospheric-lifetime than NOx (days to weeks compared with hours to days). This leads to a smaller contribution in the advection of NOx from fires occurring in north-west India during the post-monsoon. In addition, fire emissions are generated using polar orbiting satellites which have a single daytime overpass and thus may miss fires which have a short burn time; fire emissions inventories currently struggle to detect agricultural waste burning fires due to their small size and often short burn times (Zhang et al., 2020; Liu et al., 2020). Although we have used VIIRS in this study (which is able to detect smaller, lower temperature fires than MODIS) the total emissions from agricultural waste burning may still be underestimated (Zhang et al., 2020; Liu et al., 2020). With the introduction of geostationary satellites and sensors which can continuously detect smaller fires (e.g. Himawari) it should be easier to constrain the emissions from agricultural waste burning in the future.

Figure 6: What median accumulated emission values do the gray shading represent for Oct-Nov-Dec-Jan?

This is mentioned in the figure caption:

Figure 6. Wind rose of median accumulated $NO_x$ emissions (µg m$^{-2}$) from 4-day back trajectories with 6-hr time steps arriving at Delhi in 2017 and 2018 for pre-monsoon (Feb-May), monsoon (May-Oct) and post-monsoon (Oct-Feb) seasons. Total accumulated emissions are indicated by the shading of the segments. The area of the segment indicates the non-local contribution to the total integrated emissions. The percentage of local emissions is indicated on the circles. The number of days on which each wind direction occurs in each season is also indicated in brackets. For example, NW in the post-monsoon is very polluted (~350 µg m$^{-2}$), 65% of total integrated emissions are non-local and this wind direction occurs on 179 days.

Line 506-525: The paper starts (in the Introduction section) with a description of the post-monsoon agricultural burning in NW India and related aspects (without discussing the transport sector with associated statistics and known emissions). Towards the end of the paper, the authors find that the non-local transport sector dominates in NOX concentration in Delhi. Figure 4 TCNO2 anomaly plot for the post-monsoon/winter seasons showed positive anomalies along the transport pathways from agricultural burning areas in NW to Delhi. If the transport sector emission remains constant, more or less, around the year, the additional burden of NO2 may be attributed to the burning activities in the region. However, it is not captured in the sectorial analysis, indicating that the NO2 inventories in the model need to be updated.

If authors are confident and accurate about their analysis and results, they should stick to their conclusion. However, the authors should briefly discuss the strengths and weaknesses of the models and inventories employed in the study.

Regarding fire emission data, there could be possible gaps in the inventories, which are constrained by limited, once-a-day satellite observations. Bringing a high-res fire dataset with more frequent observations will help further constrain the emission inventories, as rightly stated by the authors. However, note that Himawari sees the

NW Indian region at extreme slant angles, at which the radiative transfer and detection of atmospheric/land parameters become more challenging.

We have updated several sections of the manuscript to discuss the disagreement between the accumulated satellite TCNO2 and NOx emissions back trajectories with regards to fire emissions in north-west India:

[revised manuscript text omitted]

The results section should be structured into sub-section according to the types of analysis.

We have added in separate sections for the results.

---

## Author Response (AR2)

Thank you again to the reviewers and editor for taking the time to provide their useful comments on this manuscript. We respond to the comments below in **blue** and include any changes to the manuscript in **green**.

**Editor Review**

The revised version of the manuscript has much improved as stated by the reviewers. Please address the technical correction needed mentioned by referee#2. Referee#1 emphasises (s)he still misses a consistent and adequate sectorial analysis which now doesn't accurately bring out the impact of crop burning on NO2 levels in the region, indicating that NO2 inventories should be updated in the models. This could be a good suggestion that authors can offer to the modelers. So this point could be better addressed. I would like to ask you to address this better in the next version of the manuscript.

So this refers to: "One major inconsistency noticed in the paper is that the paper starts (in the Introduction section) with a description of the post-monsoon agricultural burning in NW India and related aspects (without discussing the transport sector with associated statistics and known emissions). Towards the end of the paper, the authors find that the non-local transport sector dominates in NOx concentration in Delhi. Figure 4 TCNO2 anomaly plot for the post-monsoon/winter seasons showed positive anomalies along the transport pathways from agricultural burning areas in NW to Delhi. If the transport sector emission remains constant, more or less, around the year, the additional burden of NO2 may be attributed to the burning activities in the region. However, it is not captured in the sectorial analysis, indicating that the NO2 inventories in the model need to be updated. The authors should discuss the strengths and weaknesses of the models and inventories employed in the study.

We have added a discussion on the mismatch between the TCNO2 and NOx emissions analysis to section 4.1 lines 538-541:

It should be noted that the mismatch between the spatial pattern of  $TCNO_2$  anomalies, which clearly indicate increased  $TCNO_2$  over agricultural waste burning regions in the post-monsoon, and the  $NO_x$  emissions sectoral analysis may suggest fire emissions are underestimated in the fire emissions dataset. The reasons for this are discussed further in section 4.3.

And the discussion of the small contribution of fires and reasons for this which we already included in previous iterations of the paper has been extended:

Section 4.3, lines 560-599:

**4.3 Comparison to previous work**

The results of this study are in line with previous work by Jethva *et al.* (2018) and Sembhi *et al.* (2020). Jethva *et al.*, (2018) used 3-day HYSPLIT back trajectories which were released from 3 different altitudes (100 m, 500 m and 1500 m) in Delhi each day between October-November 2013-2016 at 13:30 local time. Trajectories were grouped according to the 24-hour averaged PM2.5 concentration at the US Embassy in Delhi (0 to <100  $\mu$ g m-3, 100 to <200  $\mu$ g m-3, 200 to <300  $\mu$ g m-3 and >300  $\mu$ g m-3). In most cases, near surface trajectories passed over crop burning regions in north-west India (Punjab and Haryana) (52%, 81%, 89% 84%, respectively). Thus, indicating air masses passing over crop burning regions are associated with increased PM2.5 concentrations in Delhi. In addition, Jethva *et al.*, (2018) estimated that trajectories took around 14-22 hours to be advected from Punjab and Haryana to Delhi indicating the potential for the advection of NOx emissions to Delhi too. Sembhi *et al.* (2020) used a model to simulate

air quality in Delhi during a poor AQ episode in 2016 with and without the implementation of the SSWA. They found that timing shift in agricultural burning in north-west India caused by the introduction of the SSWA contributed only around 3% to the poor AQ observed, indicating that this was largely driven by other factors. We also find that trajectories originating from the north-west during post-monsoon months have a polluted footprint in our analysis of satellite data and emissions. Both previous studies from Jethva et al. (2018) and Sembhi et al. (2020) suggest the potential for the advection of NOx fire emissions towards Delhi from source regions. However, within our work we do not see an impact from the advection of NOx fire emissions, which could be for several reasons. Firstly, Jethva et al. (2018) do not consider the interaction of boundary layer height and trajectory height when including trajectories in their analysis. Whereas, in this study, fire (and anthropogenic) emissions are only accumulated if the trajectory is within the boundary layer, which is very shallow during the post-monsoon. As a result, few trajectories are accumulated. Since fire emissions are buoyant and create plumes, which often extend above the boundary layer, the influence of fires may be underestimated in this study. Secondly, Sembhi et al. (2020) focussed on PM2.5, which has a much longer atmospheric-lifetime than NOx (days to weeks compared with hours to days). In our results, the shorter atmospheric lifetime of  $NO_{x}$ , relative to  $PM_{2.5}$ , leads to a smaller contribution in the advection of NOx from fires, occurring in north-west India during the post-monsoon, towards Delhi.. Finally, and arguably most importantly, fire emissions are generated using polar orbiting satellites which have a single daytime overpass and thus may miss fires which have a short burn time; fire emissions inventories currently struggle to detect agricultural waste burning fires due to their small size and often short burn times (Zhang et al., 2020; Liu et al., 2020). Although we have used VIIRS in this study (which is able to detect smaller, lower temperature fires than MODIS) the total emissions from agricultural waste burning may still be underestimated (Zhang et al., 2020; Liu et al., 2020). In addition, inventories struggle with fire detection during hazy periods, particularly those which use active fire detection (such as FINNv2.5 used in this study) leading to underestimations in fire emissions. This is supported by the large range in fire emissions estimates for November 2018, ranging from 0.63 Tg to 5.52 Tg. To accurately quantify the influence of fire emissions on Delhi AQ in the post-monsoon fire emissions inventories need to overcome these known issues. However, with the introduction of geostationary satellites and sensors which can continuously detect smaller fires (e.g. Himawari) it should be easier to constrain the emissions from agricultural waste burning in the future.

**Reviewer Report #1**

Well done to the authors for addressing all my comments. Just one pending issue - Table 1 is missing a caption.

We have added in a caption for Table 1:

Table 1. Details of the anthropogenic and fire emissions datasets used in this study. These were combined to generate daily emissions for India (and the surrounding region).